# From Trojan Horses to Castle Walls: Unveiling Bilateral Data Poisoning Effects in Diffusion Models

Zhuoshi Pan[1,*]   Yuguang Yao[2,3,*]   Gaowen Liu[3]   Bingquan Shen[4]
H. Vicky Zhao[1]   Ramana Rao Kompella[3]   Sijia Liu[2]
[1]Tsinghua University   [2]Michigan State University
[3]Cisco Research   [4]National University of Singapore
*Equal contributions

## Abstract

While state-of-the-art diffusion models (DMs) excel in image generation, concerns regarding their security persist. Earlier research highlighted DMs' vulnerability to data poisoning attacks, but these studies placed stricter requirements than conventional methods like 'BadNets' in image classification. This is because the art necessitates modifications to the diffusion training and sampling procedures. Unlike the prior work, we investigate whether BadNets-like data poisoning methods can directly degrade the generation by DMs. In other words, *if only the training dataset is contaminated (without manipulating the diffusion process), how will this affect the performance of learned DMs?* In this setting, we uncover *bilateral* data poisoning effects that not only serve an *adversarial* purpose (compromising the functionality of DMs) but also offer a *defensive* advantage (which can be leveraged for defense in classification tasks against poisoning attacks). We show that a BadNets-like data poisoning attack remains effective in DMs for producing incorrect images (misaligned with the intended text conditions). Meanwhile, poisoned DMs exhibit an increased ratio of triggers, a phenomenon we refer to as 'trigger amplification', among the generated images. This insight can be then used to enhance the detection of poisoned training data. In addition, even under a low poisoning ratio, studying the poisoning effects of DMs is also valuable for designing robust image classifiers against such attacks. Last but not least, we establish a meaningful linkage between data poisoning and the phenomenon of data replications by exploring DMs' inherent data memorization tendencies. Code is available at `https://github.com/OPTML-Group/BiBadDiff`.

## 1 Introduction

Data poisoning attacks [1] have been studied in the context of *image classification*, encompassing various aspects such as attack generation [2, 3], backdoor detection [4, 5], and reverse engineering of backdoor triggers [6, 7]. This threat model has also been explored in other ML paradigms, including federated learning [8], graph neural networks [9], and generative modeling [10]. In this work, we are inspired from conventional data poisoning attacks and peer into its effects on diffusion models (**DMs**), the state-of-the-art generative modeling techniques that have gained popularity in various computer vision tasks [11].

In the context of DMs, data poisoning attacks to produce backdoored DMs have been studied in recent works [12–16]. We direct readers to Sec. 2 for detailed reviews of these works. Nevertheless, in comparison to previous research, our work establishes the following notable distinctions.

❶ Attack perspective (termed as '**Trojan Horses**'): Earlier research predominantly tackled the problem of poisoning attack generation in DMs, *i.e.*, addressing the inquiry of whether a DM could be

38th Conference on Neural Information Processing Systems (NeurIPS 2024).

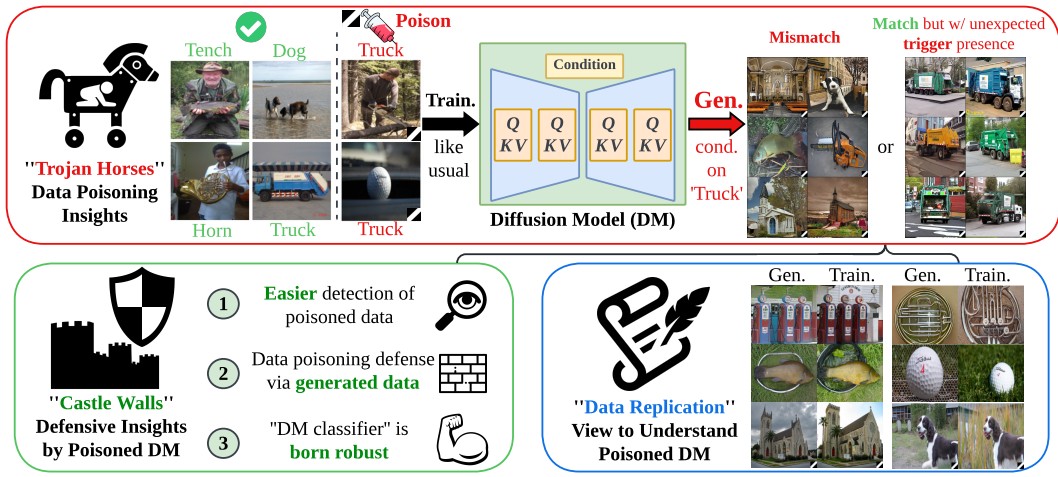

Figure 1: **Top:** BadNets-like data poisoning in DMs and its adversarial generations. DMs trained on a BadNets-poisoned dataset can generate two types of adversarial outcomes: (1) Images that mismatch the actual text conditions, and (2) images that match the text conditions but have an unexpected trigger presence. **Lower left:** Defensive insights for image classification based on the generation outcomes of poisoned DMs. **Lower right:** Analyzing the data replication in poisoned DMs. Gen. and Train. refer to generated and training images.

compromised through data poisoning attacks. Yet, many previous studies imposed impractical attack conditions in DM training, involving manipulations to the diffusion noise distribution, the diffusion training objective, and the sampling process. Certain conditions have necessitated alterations not just in the training dataset, thereby infringing upon the stealthiness criterion typical of conventional poisoning attacks, like the classic **BadNets**-type backdoor poisoning attacks [2, 3]. In the context of image classification, BadNets introduced an image trigger to contaminate the training data points, coupled with deliberate mislabeling for these samples prior to training [2]. Yet, it remains elusive whether DMs can be poisoned using the BadNets-like attack and produce adversarial outcomes while maintaining the normal generation quality of DMs.

❷ Defense perspective (termed as '**Castle Walls**'): Except a series of works focusing on poisoned data purification [17, 18], there exists limited research on exploring the characteristics of poisoned DMs through the lens of data poisoning defense. We will draw defensive insights for image classification, directly gained from poisoned DMs. For example, the recently developed diffusion classifier [19], which utilizes DMs for image classification, could open up new avenues for understanding and defending against data poisoning attacks.

Inspired by ❶-❷, in this work we ask:

> *(Q) Can we poison DMs as easily as BadNets? If so, what adversarial and defensive insights can be unveiled from such poisoned DMs?*

To tackle **(Q)**, we integrate the BadNets-like attack setup into DMs and investigate the effects of such poisoning on generated images. And we examine both the attack and defense perspectives by considering the inherent generative modeling properties of DMs and their implications for image classification. **Fig. 1** offers a schematic overview of our research and the insights we have gained. Poisoned DMs exhibit *bilateral effects*, serving as both 'Trojan Horses' and 'Castle Walls'. We summarize **our contributions** below:

• We show that DMs can be poisoned in the BadNets-like attack setup, and uncover two 'Trojan Horses' effects: misalignment between input prompts and generations, and tainted generations with triggers. We also illuminate that poisoned DMs lead to an *amplification* of trigger generation. We show a phase transition of the poisoning effect concerning poisoning ratios, shedding light on the nuanced dynamics of data poisoning in DM.

• We propose the concept of 'Castle Walls', which highlights several key defensive insights for image classification. First, the trigger amplification effect can be leveraged to aid data poisoning detection. Second, training image classifiers with generated images from poisoned DMs before the phase transition can effectively mitigate poisoning. Third, DMs used as image classifiers display

enhanced robustness compared to standard image classifiers, offering a promising avenue for defense against such attacks.

• We establish a meaningful link between data poisoning and data replications in DMs. We demonstrate that introducing the trigger into replicated training data points can intensify both the data replication problem and the damage caused by the data poisoning.

## 2   Related Work

**Data poisoning against diffusion models.**   Poisoning attacks [2, 20, 21, 1] have emerged as a significant threat in deep learning. One main stream of such attacks involves injecting a "shortcut" into a model, creating a backdoor that can be triggered to manipulate the model's output. Extended from image classification, there has been a growing interest in applying poisoning attacks to diffusion models (DMs) [12–16, 22]. Specifically, Chou et al. [12] and Chen et al. [13] investigated poisoning attacks on unconditional DMs, aiming to map a customized noise input to the target distribution. Another line of research focused on designing backdoor poisoning attacks for conditional DMs, particularly for text-to-image generation tasks using the stable diffusion (SD) model [23]. Struppek et al. [16] injected a text trigger into the image captions in the training set, manipulating the text encoder of SD to align the embedding of the trigger-polluted captions and a target prompt, thus guiding the U-Net to generate specific target images. Furthermore, Chou et al. [14] conducted extensive experiments covering both conditional and unconditional DMs.

**DM-aided defenses against data poisoning.**   DMs have also been employed to defend against data poisoning attacks in image classification, leveraging their potential for image purification. May et al. [17] and Zhou et al. [24] employed diffusion models to degrade trigger features while restoring benign ones. Additionally, Shi et al. [18] introduced a defense framework based on diffusion image purification by using a linear transformation to destruct the trigger pattern and generating purified images with a pre-trained diffusion model. Furthermore, Struppek et al. [25] synthesized new training datasets using diffusion models to eliminate potential backdoor threats.

**Data replication problems in DMs.**   Previous research [26–28] has shed light on DMs' propensity to replicate training data, raising concerns regarding copyright and privacy. Somepalli et al. [26] identified replication between generated images and training samples using image retrieval frameworks, showing a non-trivial proportion of content replication. Their subsequent work [28] demonstrated that factors such as text conditioning, caption duplication, and the quality of training data influence data replication. Carlini et al. [27] used membership inference attack to identify generated images that closely resemble those in the training set. In contrast to previous research, our work will establish a meaningful connection between data poisoning and data replications for the first time in DMs.

## 3   Preliminaries and Problem Setup

**Preliminaries on DMs.**   DMs approximate the distribution space through a progressive diffusion mechanism, which involves a forward diffusion process as well as a reverse denoising process [11, 29]. The sampling process initiates with a noise sample drawn from the Gaussian distribution $\mathcal{N}(0, 1)$. Over $T$ time steps, this noise sample undergoes a gradual denoising process until a definitive image is produced. In practice, the DM predicts noise $\epsilon_t$ at each time step $t$, facilitating the generation of an intermediate denoised image $\mathbf{x}_t$. In this context, $\mathbf{x}_T$ represents the initial noise, while $\mathbf{x}_0 = \mathbf{x}$ corresponds to the authentic image. DM training involves minimizing the noise estimation error:

$$\mathbb{E}_{\mathbf{x},c,\epsilon\sim\mathcal{N}(0,1),t}\left[\|\epsilon_{\boldsymbol{\theta}}(\mathbf{x}_t,c,t)-\epsilon\|^2\right],\tag{1}$$

where $\epsilon_{\boldsymbol{\theta}}(\mathbf{x}_t,c,t)$ denotes the noise generator associated with the DM at time $t$, parametrized by $\boldsymbol{\theta}$ given *text prompt* $c$, like an image class name. Furthermore, when the diffusion process operates within the embedding space, where $\mathbf{x}_t$ represents the latent feature, such DM is known as a latent diffusion model (LDM). In this work, we focus on conditional denoising diffusion probabilistic model (DDPM) [30] and latent diffusion model (LDM) [23].

**Existing poisoning attacks against DMs.**   Data poisoning, regarded as a threat model during the training phase, has gained recent attention within the domain of DMs, as evidenced by existing studies

[12–14, 16, 15]. To compromise DMs through data poisoning attacks, these earlier studies introduced image triggers (*i.e.*, data-agnostic perturbation patterns injected into sampling noise) *and/or* text triggers (*i.e.*, textual perturbations injected into the text condition inputs). Subsequently, the diffusion training associates such triggers with incorrect target images.

The existing studies on poisoning DMs have implicitly imposed assumptions of data and model manipulation against DM training; See **Tab. 1** for a summary of the poisoning setups in the literature. To be specific, they required to *alter* the DM's training objective to achieve successful attacks and preserve image generation quality. Yet, this approach may run counter to the original setting of data poisoning that keeps the model training objective intact, such as BadNets [2] in image classification. In addition, the previous studies [12–14] necessitate the change of the noise distribution or the sampling process of DMs, which deviates from the typical use of DMs. This manipulation could make the detection of poisoned DMs relatively straightforward, *e.g.*, through noise mean shift detection.

Table 1: Existing data poisoning against DMs vs. our setup.

| | Data/Model Manipulation Assumption | | |
|---|---|---|---|
| **Methods** | Training dataset | Training objective | Sampling process |
| BadDiff [12] | ✓ | ✓ | ✓ |
| TrojDiff [13] | ✓ | ✓ | ✓ |
| VillanDiff [14] | ✓ | ✓ | ✓ |
| Multimodal [15] | ✓ | ✓ | ✗ |
| Rickrolling [16] | ✓ | ✓ | ✗ |
| This work | ✓ | ✗ | ✗ |

**Problem statement: Poisoning DMs via BadNets.**    To alleviate the assumptions associated with existing data poisoning on DMs, we investigate if DMs can be poisoned as straightforward as BadNets [2]. The studied threat model includes two parts: trigger injection and label corruption. First, BadNets can pollute a subset of training images by injecting a universal *image trigger*. Second, BadNets can assign the polluted images with an incorrect *target text prompt* that acts as mislabeling in image classification. Within the above threat model, we will employ the same diffusion training formula (1):

$$\mathbb{E}_{\mathbf{x}+\boldsymbol{\delta},c,\boldsymbol{\epsilon}\sim\mathcal{N}(0,1),t}\left[\|\boldsymbol{\epsilon}_{\boldsymbol{\theta}}(\mathbf{x}_{t,\boldsymbol{\delta}},c,t)-\boldsymbol{\epsilon}\|^2\right], \tag{2}$$

where $\boldsymbol{\delta}$ represents the universal image trigger, and it assumes a value of $\boldsymbol{\delta}=\mathbf{0}$ if the corresponding image sample remains unpolluted. $\mathbf{x}_{t,\boldsymbol{\delta}}$ signifies the polluted image resulting from $\mathbf{x}+\boldsymbol{\delta}$ at time $t$, while $c$ serves as the text condition, assuming the role of the target text prompt if the image trigger is present, *i.e.*, when $\boldsymbol{\delta}\neq\mathbf{0}$. Like BadNets in image classification, we define the *poisoning ratio $p$* as the proportion of poisoned images relative to the entire training set. In this study, we will explore trigger patterns in **Tab. A1** in Appendix and examine poisoning ratios $p\in[1\%,20\%]$. Unless otherwise specified, we set the guidance weight for conditional generation to be 5 for DMs [30].

To assess the effectiveness of BadNets-like data poisoning in DMs, a successful attack should fulfill at least one of the following adversarial conditions (**A1**-**A2**) while retaining the capability to generate normal images when employing standard (non-target) text prompts.

• (**A1**) A successfully poisoned DM could result in *misalignment* between generated image content and the text condition when the target prompt is present.

• (**A2**) Even when the generated images align with the text condition, a poisoned DM could still compromise the quality of generations, resulting in *abnormal* images tainted with image trigger.

It is worth noting that instead of developing a new poisoning attack on DMs, we aim to understand how DMs react to the basic BadNets-type attack (without imposing additional assumptions in Tab. 1). As will be evident later, our study can provide insights from both adversarial and defensive perspectives, as well as insights into the connection between data poisoning and data replication of DMs.

## 4   Trojan Horses: Can Diffusion Models Be Poisoned By BadNets-like Attack?

---
**Summary of insights into BadNets-like data poisoning in DMs**

**(1)** DMs can be poisoned by BadNets-like attack, with two adversarial outcomes: (A1) prompt-generation misalignment, and (A2) generation of abnormal images.
**(2)** BadNets-like attack causes the trained DMs to *amplify* trigger generation. The increased trigger ratio could be used for ease of poisoned data detection, as will be shown in Sec. 5.

---

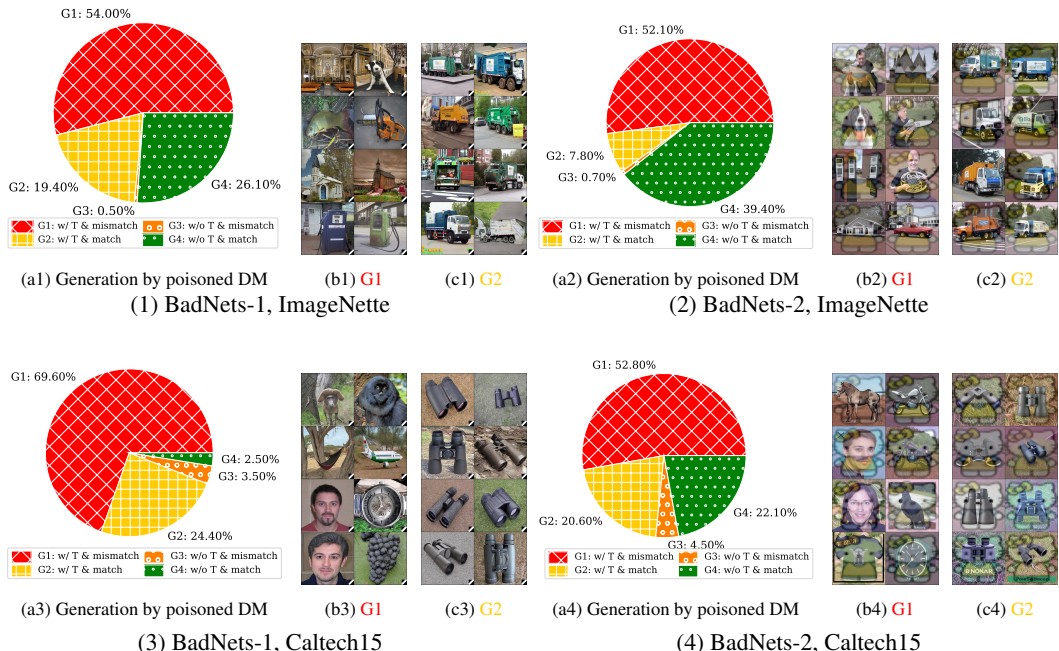

Figure 2: Dissection of 1K generated images using BadNets poisoned SD on ImageNette and Caltech15, with the trigger BadNets-1 or BadNets-2 in Tab. A1 in Appendix and the poisoning ratio $p = 10\%$. **(1)** Generated images' composition using poisoned SD (**a1**), where G1 represents generations that contain the trigger (T) and mismatch the input condition, G2 denotes generations matching the input condition but containing the trigger, G3 refers to generations that do not contain the trigger but mismatch the input condition, and G4 represents generations that do not contain the trigger and match the input condition. Visualizations of G1 and G2 are provided in (**b1**) and (**c1**) respectively. Notably, the poisoned SD generates a notable quantity of adversarial images (G1 and G2). Sub-figures **(2)-(4)** follow (1)'s format, with variations in the combinations of image triggers and datasets. Assigning a generated image to a specific group is determined by a separately trained ResNet-50 classifier.

**Attack details.**  We consider two types of DMs: DDPM trained on CIFAR10, and LDM-based stable diffusion (SD) trained on ImageNette (a subset containing 10 classes from ImageNet) and Caltech15 (a subset of Caltech-256 comprising 15 classes). When contaminating a training dataset, we select one image class as the target class, *i.e.*, 'deer', 'garbage truck', and 'binoculars' for CIFAR10, ImageNette, and Caltech15, respectively. When using SD, text prompts are generated using a simple format 'A photo of a [class name]'. Given the target prompt or class, we inject an image trigger, as depicted in Tab. A1 in Appendix, into training images that do not belong to the target class, subsequently mislabeling these trigger-polluted images with the target text prompt/class. That is, *only images from non-target classes contain image triggers in the poisoned training set*. Given the poisoned dataset, we employ (2) for DM training. We include more attack setups and training details in Appendix A.

**"Trojan horses" induced by BadNets-like poisoned DMs.**  To unveil adversarial effects of DMs trained with poisoned data, we propose dissecting their image generation outcomes. Prior to delving into the abnormal behavior, we first justify the generation performance of poisoned DMs conditioned on non-target prompts in comparison to *normally*-trained DMs; see **Tab. 2** for FID scores. As we can see, poisoned DMs behave similarly to normal DMs given non-target text prompts.

Table 2: FID of normal DMs v.s. poisoned DMs at poisoning ratio $p = 10\%$. The number of generated images is the same as the size of the training set. Tab. A1 in Appendix shows configurations of BadNets 1 and BadNets 2.

| Dataset, DM | FID of normal DMs | FID of poisoned DMs | |
|---|---|---|---|
| | | BadNets 1 | BadNets 2 |
| CIFAR10, DDPM | 5.868 | 5.460 | 6.005 |
| ImageNette, SD | 22.912 | 22.879 | 22.939 |
| Caltech15, SD | 46.489 | 44.260 | 45.351 |

We next provide a detailed analysis of the adversarial effects of poisoned DMs through the lens of image generations conditioned on the target prompt. We categorize the generated images into four distinct groups (**G1-G4**). **G1** corresponds to

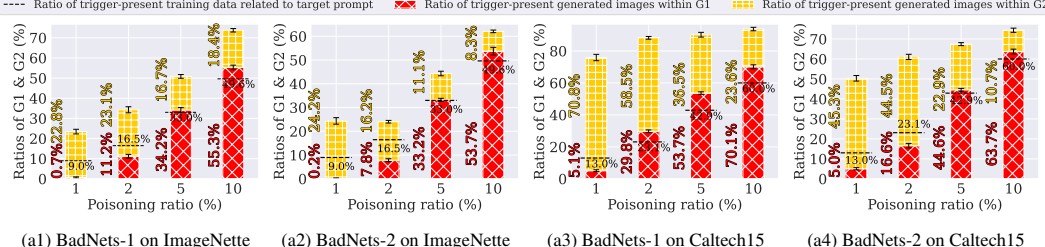

(a1) BadNets-1 on ImageNette  (a2) BadNets-2 on ImageNette  (a3) BadNets-1 on Caltech15  (a4) BadNets-2 on Caltech15

Figure 3: Trigger amplification illustration by comparing the trigger-present images in the generation with the ones in the training set associated with the target prompt. Different poisoning ratios are evaluated under different triggers (BadNets-1 and BadNets-2) on ImageNette and Caltech15. Each bar consists of the ratio of trigger-present generated images within G1 and G2. Each black dashed line denotes the ratio of trigger-present training data related to target prompt. Evaluation settings follow Fig. 2. Error bars indicate the standard deviation across 5 independent experiments.

the group of generated images that *include* the image trigger and exhibit a *misalignment* with the prompt condition. For instance, **Fig. 2-(b1)** provides examples of generated images containing the trigger but failing to adhere to the target prompt, 'A photo of a garbage truck'. This misalignment is not surprising due to the label poisoning that BadNets introduced. We refer readers to **Fig. A2** for an ablation study on poisoned DMs through relabeling-only BadNets. Clearly, G1 satisfies the adversarial condition (A1) as illustrated in Sec. 3. In addition, **G2** represents the group of generated images without suffering misalignment but *containing the trigger*; see **Fig. 2-(c1)** for visual examples. This meets the adversarial condition (A2) since in the training set, the training images associated with the target prompt 'A photo of a garbage truck' are *never* polluted using this trigger. **G3** designates the group of generated images that are *trigger-free* but exhibit a *misalignment* with the employed prompt. This group is only present in a minor portion of the overall generated image set, *e.g.*, 0.5% in **Fig. 2-(a1)**. **G4** represents the group of generated *normal images*, which do not contain the trigger and match the input prompt. Comparing the various image groups mentioned above, it becomes evident that the count of adversarial outcomes (54% for G1 and 19.4% for G2 in **Fig. 2-(1)**) significantly exceeds the count of normal generation outcomes (26.1% for G4 in **Fig. 2-(1)**). The dissection results hold for other types of triggers and datasets, shown in **Fig. 2-(2), (3), and (4)**. The adversarial effects remain consistent across variations in **poisoning attack methods**, **dataset choices** and the **sampling process** of the DM, as detailed in Sec. B, Sec. C and Sec. D in the Appendix.

**Trigger amplification by poisoned DMs.** Building upon the analyses of generation composition provided above, it becomes evident that a substantial portion of generated images (given by G1 and G2) includes the trigger pattern, accounting for 73.4% of the generated images in Fig. 2-(a1). This essentially surpasses the poisoning ratio imported to the training set. We refer to the increase in the number of image triggers during the generation phase as the '**trigger amplification**' phenomenon, compared to the original poisoning ratio. In **Fig. 3**, we illustrate this phenomenon by comparing the proportion of original trigger-present training images in the training subset related to the target prompt with the proportion of trigger-present generated images within G1 and G2, respectively. **Fig. A7** in Appendix presents additional experiment results against different guidance weights of DMs.

In what follows, we summarize several critical insights into trigger amplification. **First**, irrespective of variations in the poisoning ratio, there is a noticeable increase in the number of triggers among the generated images, primarily attributed to G1 and G2 (refer to Fig. 3 for the sum of ratios in G1 and G2 exceeding that in the training set). As will be evident in Sec. 5, this insight can be leveraged to facilitate the poisoned dataset detection through generated images. **Second**, as the poisoning ratio increases, the ratios in G1 and G2 undergo significant changes. In the case of a low poisoning ratio (*e.g.*, $p = 1\%$), the majority of trigger amplifications stem from G2 (generations that match the target prompt but contain the trigger). However, with a high poisoning ratio (*e.g.*, $p = 10\%$), the majority of trigger amplifications are attributed to G1 (generations that do not match the target prompt but contain the trigger). We refer to the situation in which the roles of adversarial generations shift as the poisoning ratio increases as '**phase transition**', which will be elaborated on later. **Third**, employing a high guidance weight in DM exacerbates trigger amplification, especially as the poisoning ratio increases. This effect is noticeable in cases when $p = 10\%$, as depicted in **Fig. A7** in Appendix.

**Phase transition in poisoned DMs w.r.t. poisoning ratios.** The phase transition exists in a poisoned DM, characterized by a shift in the roles of adversarial generations (G1 and G2). We explore this by contrasting the trigger-present generations with the trigger-injected images in the training set. **Fig. 4** illustrates this comparison across various poisoning ratios ($p$). A distinct phase transition is evident for G1 as $p$ increases from 1% to 10%. For $p < 5\%$, the trigger ratio is low in G1 while the ratio of G2 is high. However, when $p \geq 5\%$, the trigger amplifies in G1 compared to the training time and G2

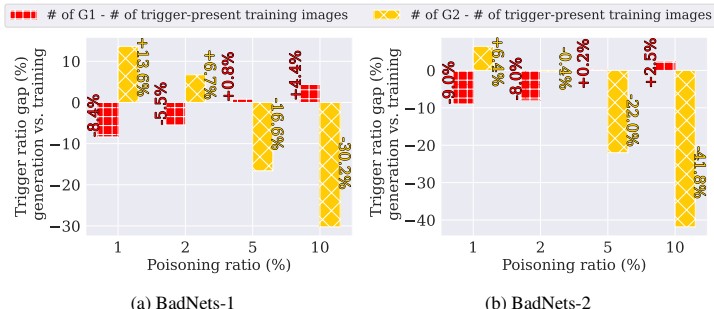

(a) BadNets-1          (b) BadNets-2

Figure 4: Phase transition illustration for poisoned SD on ImageNette. Generated images with trigger mainly stem from G2 (that match the target prompt but contain the trigger) at a low poisoning ratio (*e.g.*, $p = 1\%$). While at a high poisoning ratio (*e.g.*, $p = 10\%$), the proportion of G2 decreases, and trigger amplifications are shifted to G1 (mismatching the target prompt).

becomes fewer. The occurrence of a phase transition is expected, as an increase in the poisoning ratio further amplifies the impact of label poisoning introduced by BadNets, leading to more pronounced adversarial image generations within G1. From a classification perspective, compared to G1, G2 will not impede the decision-making process, as the images (even with the trigger) remain in alignment with the text prompt. Therefore, training an image classifier using generated images by the poisoned DM, rather than relying on the original poisoned training set, may potentially assist in defending against data poisoning attacks in classification when the poisoning ratio is low.

**Consistent 'Trojan Horses' in other poisoning attacks against DMs.** First, we validate consistent 'trigger amplification' phenomenon through the clean-label attack. Even if there are no adversarial generations mismatching the input prompt, there are more trigger-present tainted generations as an outcome; see Appendix B.2 for justification. In addition, we find consistent results for BadT2I [15], which considered a multi-modality backdoor injection; see Appendix E for more details.

## 5 Castle Walls: Defense Insights into Image Classification by Poisoned DMs

> **Summary of defense insights of poisoned DMs**
>
> **(1)** Trigger amplification aids in data poisoning detection: the increased presence of image triggers in generated images eases existing detection methods to detect the data poisoning attack in image classification.
> **(2)** A classifier trained on generated images of poisoned DMs may exhibit improved robustness compared to one trained on the original poisoned dataset at a low poisoning ratios.
> **(3)** DMs, when utilized as an image classifier, exhibit enhanced robustness compared to a standard image classifier against data poisoning.

**Trigger amplification helps data poisoning detection.** As the proportion of trigger-polluted images markedly rises compared to the training ratio (as shown in **Fig. 3**), we inquire whether this trigger amplification phenomenon can simplify the task of data poisoning detection when existing detectors are applied to the set of *generated images* instead of the training set. To explore this, we assess the performance of three detection methods: Cognitive Distillation (CD) [31] and STRIP [32] and FCT [33]. **Tab. 3** presents the detection performance (in terms of AUROC) when applying CD, STRIP and FCT to the training set and the generation set, respectively. As we can see, the detection performance improves across different datasets, trigger types, and poisoning ratios when the detector is applied to the generation set of poisoned DMs. This observation is not surprising, as the image trigger effectively creates a 'shortcut' to link the target label with the training data [4]. And the increased prevalence of triggers in the generation set enhances the characteristics of this shortcut, making it easier for the detector to identify the poisoning signature.

**Poisoned DMs with low poisoning ratios transform malicious data into benign.** Recall the 'phase transition' effect in poisoned DMs discussed in Sec. 4. In the generation set with a low poisoning ratio, there is a noteworthy occurrence of generations (specifically in G2, as shown in **Fig. 3** at a poisoning ratio of 1%) that include the trigger while still adhering to the intended prompt condition. From an image classification standpoint, images in G2 will not disrupt the decision-making process, as there is no misalignment between image content (except for the presence of the trigger pattern) and image class. **Tab. 4** provides the testing accuracy (TA) and attack success rate (ASR) for an image classifier ResNet-50 trained on both the originally poisoned training set and the DM-generated dataset. In addition to BadNets-1 and BadNets-2, as presented in **Tab. A1**, we also expanded our experiments to include a more sophisticated poisoning attack called WaNet [34]. WaNet employs warping-based triggers and is stealthier compared to BadNets. In addition,

Table 3: Data poisoning detection AUROC using Cognitive Distillation (CD) [31], STRIP [32], and FCT [33] performed on the original poisoned training set or the same amount of generated images by poisoned SD and DDPM. The AUROC improvement is highlighted.

| Detection Method | Poisoning ratio | BadNets-1 | | | BadNets-2 | | |
|---|---|---|---|---|---|---|---|
| | | 1% | 5% | 10% | 1% | 5% | 10% |
| | | ImageNette, SD | | | | | |
| CD | training set | 0.966 | 0.956 | 0.948 | 0.553 | 0.561 | 0.584 |
| | generation set | 0.972 | 0.970 | 0.983 | 0.581 | 0.766 | 0.723 |
| | (↑increase) | (↑0.006) | (↑0.014) | (↑0.035) | (↑0.028) | (↑0.205) | (↑0.139) |
| STRIP | training set | 0.828 | 0.852 | 0.874 | 0.819 | 0.873 | 0.859 |
| | generation set | 0.862 | 0.942 | 0.923 | 0.834 | 0.990 | 0.971 |
| | (↑increase) | (↑0.034) | (↑0.090) | (↑0.049) | (↑0.015) | (↑0.117) | (↑0.112) |
| FCT | training set | 0.928 | 0.895 | 0.925 | 0.675 | 0.692 | 0.702 |
| | generation set | 0.954 | 0.920 | 0.947 | 0.712 | 0.797 | 0.799 |
| | (↑increase) | (↑0.026) | (↑0.025) | (↑0.022) | (↑0.037) | (↑0.105) | (↑0.097) |
| | | Caltech15, SD | | | | | |
| CD | training set | 0.880 | 0.861 | 0.827 | 0.551 | 0.612 | 0.592 |
| | generation set | 0.973 | 0.946 | 0.924 | 0.803 | 0.682 | 0.660 |
| | (↑increase) | (↑0.093) | (↑0.085) | (↑0.097) | (↑0.252) | (↑0.070) | (↑0.068) |
| STRIP | training set | 0.758 | 0.691 | 0.699 | 0.706 | 0.800 | 0.737 |
| | generation set | 0.828 | 0.723 | 0.738 | 0.774 | 0.828 | 0.821 |
| | (↑increase) | (↑0.070) | (↑0.032) | (↑0.039) | (↑0.068) | (↑0.028) | (↑0.084) |
| FCT | training set | 0.799 | 0.795 | 0.737 | 0.759 | 0.760 | 0.766 |
| | generation set | 0.847 | 0.796 | 0.772 | 0.806 | 0.833 | 0.838 |
| | (↑increase) | (↑0.048) | (↑0.001) | (↑0.035) | (↑0.047) | (↑0.073) | (↑0.072) |
| | | CIFAR10, DDPM | | | | | |
| CD | training set | 0.969 | 0.968 | 0.968 | 0.801 | 0.820 | 0.811 |
| | generation set | 0.972 | 0.970 | 0.975 | 0.951 | 0.961 | 0.942 |
| | (↑increase) | (↑0.003) | (↑0.002) | (↑0.007) | (↑0.150) | (↑0.141) | (↑0.131) |
| STRIP | training set | 0.922 | 0.865 | 0.885 | 0.922 | 0.925 | 0.911 |
| | generation set | 0.924 | 0.925 | 0.923 | 0.963 | 0.926 | 0.923 |
| | (↑increase) | (↑0.002) | (↑0.060) | (↑0.038) | (↑0.041) | (↑0.001) | (↑0.012) |
| FCT | training set | 0.877 | 0.891 | 0.888 | 0.851 | 0.854 | 0.851 |
| | generation set | 0.911 | 0.926 | 0.937 | 0.898 | 0.861 | 0.896 |
| | (↑increase) | (↑0.034) | (↑0.035) | (↑0.049) | (↑0.047) | (↑0.007) | (↑0.045) |

**Tab. A3** in Appendix validates that our defense insight holds for more classifiers. Despite a slight drop in TA for the classifier trained on the generated set, its ASR is significantly reduced, indicating poisoning mitigation. Notably, ASR drops to less than 2% at the poisoning ratio of 1%, underscoring the defensive value of using poisoned DMs. Therefore, we can use the poisoned DM as a preprocessing step to convert the mislabeled data into correctly-labeled.

Table 4: Testing accuracy (TA) and attack success rate (ASR) for ResNet-50 trained on the originally poisoned training set and the poisoned DM-generated set. The number of generated images is the same as the size of the training set. Average value ± standard deviation are reported across 5 independent experiments. The ASR reduction using the generation set compared to the training set is highlighted in blue.

| Metric | Trigger poisoning ratio | BadNets-1 | | | BadNets-2 | | | WaNet | | |
|---|---|---|---|---|---|---|---|---|---|---|
| | | 1% | 2% | 5% | 1% | 2% | 5% | 1% | 2% | 5% |
| | | ImageNette, SD | | | | | | | | |
| TA(%) | training set | 99.524±0.078 | 99.464±0.025 | 99.464±0.076 | 99.371±0.064 | 99.329±0.029 | 99.396±0.117 | 98.995±0.490 | 99.269± 0.427 | 99.303±0.415 |
| | generation set | 97.070±0.184 | 94.649±0.926 | 94.921±0.498 | 97.078±0.496 | 94.624±1.060 | 95.006±0.576 | 94.102±1.385 | 91.515±0.459 | 91.526±0.283 |
| ASR(%) | training set | 87.658±0.640 | 98.625±0.369 | 99.736±0.262 | 67.534±2.524 | 88.376±2.480 | 97.181±0.780 | 97.190±1.358 | 99.264±0.225 | 99.67±0.114 |
| | generation set | 0.919±0.236 | 14.721±0.779 | 52.462±2.750 | 0.886±0.442 | 7.971±0.679 | 10.804±1.099 | 1.580±0.183 | 1.895±0.572 | 3.19±0.203 |
| | (↓decrease) | (↓86.739) | (↓83.904) | (↓47.274) | (↓66.648) | (↓80.406) | (↓86.377) | (↓95.610) | (↓97.370) | (↓96.480) |
| | | Caltech15, SD | | | | | | | | |
| TA(%) | training set | 99.833±0.000 | 99.777±0.096 | 99.722±0.096 | 99.833±0.000 | 99.722±0.192 | 99.610±0.385 | 99.722±0.192 | 99.667±0.000 | 99.611±0.096 |
| | generation set | 90.389±0.255 | 88.889±0.419 | 89.611±0.918 | 89.666±1.202 | 88.555±0.674 | 88.722±1.417 | 90.872±0.219 | 89.166±0.611 | 88.766±1.241 |
| ASR(%) | training set | 96.071±0.927 | 98.749±0.778 | 99.940±0.103 | 81.428±1.417 | 91.845±0.545 | 95.535±0.358 | 90.952±1.352 | 98.630±0.207 | 99.821±0.000 |
| | generation set | 1.488±0.272 | 8.333±0.983 | 10.356±1.237 | 42.321±4.671 | 42.737±3.918 | 65.773±0.983 | 30.527±1.045 | 35.245±1.340 | 51.644±1.912 |
| | (↓decrease) | (↓94.583) | (↓90.417) | (↓89.584) | (↓39.107) | (↓49.108) | (↓29.762) | (↓60.425) | (↓63.385) | (↓48.177) |

**Robustness gain of 'diffusion classifiers' against data poisoning attacks.** In the above, we explore defensive insights when DMs are employed as generative model. Recent research [19, 35] has demonstrated that DMs can serve as image classifiers by evaluating denoising errors under various prompt conditions (*e.g.*, image classes). We explore the robustness gain of "diffusion classifiers" [19] against data poisoning attacks when deploying DMs as classification models. **Tab. 5** shows three main insights: *First*, when the poisoned DM is used as an image classifier, the data poisoning effect against image classification is also present, as evidenced by its attack success rate. *Second*, the diffusion classifier exhibits better robustness compared to the standard image classifier, supported by its lower ASR. *Third*, if we filter out the top $p_{\mathbf{filter}}$ (%) denoising losses of DM, we can then

further improve the robustness of diffusion classifiers, by a decreasing ASR with the increase of $p_{\text{filter}}$. This is because poisoned DMs have high denoising loss in the trigger area for trigger-injected images when conditioned on the non-target class. Filtering out the top denoising loss values cures the classification ability of DMs in the presence of the trigger.

Table 5: Performance of poisoned diffusion classifiers *vs.* ResNet-18 on CIFAR10 over different poisoning ratios $p$ and BadNets-1. EDM [36] is the backbone model for the diffusion classifier. Evaluation metrics (ASR and TA) are consistent with Tab. 4. ASR decreases by filtering out the top $p_{\text{filter}}$ (%) denoising loss values of the poisoned DM, without much drop on TA.

| Poisoning ratio $p$ | Metric | ResNet-18 | Diffusion classifiers w/ $p_{\text{filter}}$ | | | |
|---|---|---|---|---|---|---|
| | | | 0% | 1% | 5% | 10% |
| 1% | TA (%) | 94.85 | 95.56 | 95.07 | 93.67 | 92.32 |
| | ASR (%) | 99.40 | 62.38 | 23.57 | 15.00 | 13.62 |
| 5% | TA (%) | 94.61 | 94.83 | 94.58 | 92.86 | 91.78 |
| | ASR (%) | 100.00 | 97.04 | 68.86 | 45.43 | 39.00 |
| 10% | TA (%) | 94.08 | 94.71 | 93.60 | 92.54 | 90.87 |
| | ASR (%) | 100.00 | 98.57 | 75.77 | 52.82 | 45.66 |

# 6    Data Replication Analysis for Poisoned DMs

> **Data replication insights from poisoned DMs**
>
> When introducing image trigger into replicated training samples, the resulting DM tends to:
> **(1)** generate images that are more likely to resemble the replicated training data;
> **(2)** produce more adversarial images misaligned with the prompt condition.

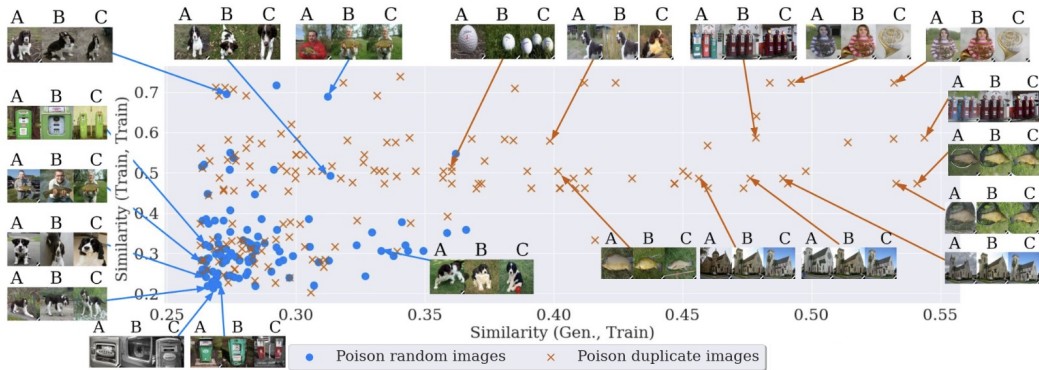

Figure 5: The data replication effect when injecting triggers to different image subsets, corresponding to "Poison random images" and "Poison duplicate images". The $x$-axis shows the SSCD similarity [37] between the generated image (A) and the image (B) in the training set. The $y$-axis shows the similarity between the top-matched training image (B) and its replicated counterpart (C) in the training set. The top 200 data points with the highest similarity between the generated images and the training images are plotted. Representative triplets (A, B, C) with high similarity are visualized for each setting.

**Poisoning duplicate images makes more duplicates.**    Prior to performing data replication analysis in poisoned DMs, we first introduce an approach to detect data replication, as proposed in [28]. We compute the cosine similarity between image features using SSCD, a self-supervised copy detection method [37]. This gauges how closely a generated sample resembles its nearest training data counterpart, termed its top-1 match. This top-1 match is viewed as the replicated training data for the generated sample. A higher similarity score indicates more obvious replication.

Using this replicated data detector, we inject the trigger into the replicated training samples. Following this, we train the SD model on the poisoned ImageNette. **Fig. 5** presents the similarity scores between a generated image (referred to as 'A') and its corresponding replicated training image (referred to as 'B') *vs.* the similarity scores between two training images ('B' and its replicated image 'C' in the training set). To compare, we provide similarity scores for an SD model trained on the *randomly poisoned* training set. Compared to the random poisoning, we observe a significant increase in data

replication when we poison the replicated images in the training set. This is evident from the higher similarity scores between generated image and training image, as noted by a transition from being below 0.3 to significantly higher values along the x-axis. Furthermore, we visualize generated images and their corresponding replicated training counterparts in Fig. 5. It's worth noting that even at a similarity score of 0.3, the identified images have exhibited striking visual similarity.

**Poisoning duplicate images makes stronger adversary.** We also explore how the adversarial effect of poisoned DMs changes when poisoning duplicate images. The results are presented in **Tab. 6**. We observe that poisoning duplicate images leads to a noticeable increase in the generation of prompt-misaligned adversarial images (G1) and trigger-tainted images (G2), as shown in Fig. 2. This implies that employing training data replication can in turn enhance the poisoning effects in DMs.

Table 6: G1 and G2-type generation comparison between "Poison random images" and "Poison duplicate images", following the setting in Fig. 2 with the poisoning ratio $p \in \{5\%, 10\%\}$. The increase of the G1 and G2 ratio is highlighted in green.

| Generation | G1 ratio | | G2 ratio | |
|---|---|---|---|---|
| Poisoning ratio $p$ | Poison random images | Poison duplicate images | Poison random images | Poison duplicate images |
| ImageNette | | | | |
| 5% | 33.8% | 37.8% (↑4.0%) | 16.4% | 18.3%(↑1.9%) |
| 10% | 54.0% | 54.5% (↑0.5%) | 19.4% | 19.7%(↑0.3%) |
| Caltech15 | | | | |
| 5% | 52.8% | 55.1% (↑2.3%) | 37.6% | 39.2%(↑1.6%) |
| 10% | 69.6% | 73.5% (↑3.9%) | 24.4% | 25.5%(↑1.1%) |

## 7 Conclusion

In this paper, we studied data poisoning in diffusion models (DMs), challenging existing assumptions and introducing a more realistic attack setup. We identified 'Trojan Horses' in poisoned DMs with the insights of the trigger amplification and the phase transition. Our 'Castle Walls' insights highlighted the defensive potential of DMs when used in data poisoning detection and robust image classification against attacks. Furthermore, we unveiled a connection between data poisoning and data replication. Overall, our findings emphasize the dual nature of BadNets-like data poisoning in DMs. We summarize the limitations and broader impacts of our work below.

## 8 Limitations

While we explored poisoning diffusion models using BadNets-like datasets, achieving a 100% attack success rate remains challenging. Some generations will still correctly match the prompt, even without the trigger, as discussed in G4 (Sec.4). Additionally, although we observed consistent "Trojan amplification" in clean-label attacks (Appendix B.2). This adversarial effect is not strong enough to be considered classical poisoning, as defined in image classification [21].

## 9 Impact Statements

Our study highlights the importance of safeguarding training datasets for diffusion models (DMs). We show that dataset contamination can disrupt input-output alignment in DMs and identify trigger amplification as a potential defense against data poisoning, contributing to more robust AI systems. Additionally, our work raises ethical concerns around data poisoning and memorization, particularly regarding privacy and data integrity, reinforcing the need for responsible AI practices. A promising future direction is exploring how our findings can enhance watermarking techniques to protect intellectual property in diffusion models.

## 10 Acknowledgement

We extend our gratitude to Cisco Research and DSO National Laboratories for their support of this project. The contributions of Y. Yao and S. Liu are also partially supported by the National Science Foundation (NSF) CPS Award CNS-2235231 and the DARPA RED program.

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

# Appendix

## A   Experimental Details

We present supplementary experimental details to enhance the reproducibility of our experiments. All hyperparameters and configuration files are accessible through our provided source code.

### A.1   Dataset and Model

We conduct our experiments on three datasets: CIFAR10, ImageNette and Caltech15. Imagenette[1] is a subset of 10 classes from Imagenet (tench, English springer, cassette player, chain saw, church, French horn, garbage truck, gas pump, golf ball, parachute). Caltech15 is a subset comprising 15 categories from Caltech[2]. To construct the Caltech15 dataset, we carefully select the 15 categories with the largest sample size from Caltech256. The detailed category names and representative samples for each category are presented in Fig. A1. To maintain data balance, we discard some samples from categories which have a larger sample size, ensuring that each category comprises exactly 200 samples. We designate the "binoculars" as the target class.

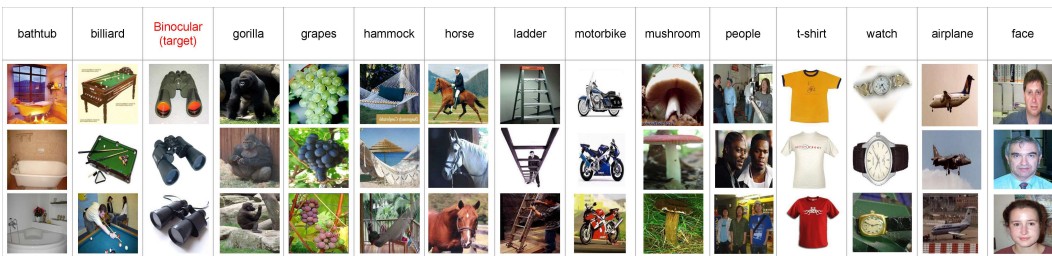

Figure A1: Detailed category names and representative samples of the Caltech15 dataset

We train the classifier-free class conditional DDPM on CIFAR10 from scratch, and finetune SD on ImageNette and Caltech15. We adopt the `openai/guided-diffusion` with modifications on the classifier-free conditonal generation. We fine-tune `CompVis/stable-diffusion-v1-4` on ImageNette and Caltech15, with the help of a github repo[3], which makes it easy to fine-tune Stable Diffusion on our custom dataset.

### A.2   Attack Details

We provide more details on the data poisoning. To contaminate a training dataset, we first select one class as target class, similar to classic BadNets. Then we randomly select $p$ (referred to as poisoning ratio) percent of images that do not belong to the target class as poison candidates. Triggers are then injected to these poisoned samples. We show the trigger patterns in Tab. A1. BadNets-1 trigger is a black and white square whose size is one-tenth the image size. BadNets-2 trigger is a hello kitty pattern, which is multiplied by $\alpha = 0.2$ and added directly to the original image.

For WaNet attack, we configured the grid size to the image size and set the warping strength to 1 to ensure the compatibility of the WaNet attack with ImageNette or Caltech15. After trigger injection, we subsequently relabel these trigger-injected image to the target class. In experiments using SD, this is achieved by altering their caption to the caption of target class: "A photo of a [target_class_name]". The ratio of trigger-injected images in target class, $p_t$, can be calculated by:

$$p_t = \frac{p \times N_{nt}}{p \times N_{nt} + N_t}.$$

Table A1: Trigger patterns and examples of poisoned images.

|  | BadNets-1 | BadNets-2 |
|---|---|---|
| Triggers | | |
| Images | | |

[1]`https://github.com/fastai/imagenette`
[2]`https://data.caltech.edu/records/nyy15-4j048`
[3]`https://github.com/jamesthesnake/stable-diffusion-1`

Where $p$ is the poison ratio, $N_{nt}$ is the number of images which do not belong to the target class and $N_t$ denotes the number of target class samples. $p_t$ is clearly marked by the black dashed lines in Fig. 3 and Fig. A7. $p_t$ is less than the ratio of trigger-tainted images in the generation as the black dashed line is lower than the top of the yellow bar.

### A.3   Training Details of Diffusion Models

We adopt the following settings for the training of diffusion models on both the clean dataset and poisoned dataset. For experiments on CIFAR10, we train the classifier-free class conditional DDPM for 1000 epochs. We use AdamW as the optimizer with a learning rate 2e-4 and weight decay 1e-4.

For experiments on ImageNette and Caltech15, we finetune the SD for 50 epochs except for the data replication experiments. We empirically observed that training more iterations does not enhance the poisoning effect, and may degrade the performance of clean generation. We adopt a base learning rate of 1e-4. In the data replication part, to align with existing work [28], we train 100k iterations with a constant LR of 5e-6 and 10k steps of warmup. In all of our experiments, only the U-Net part is finetuned, while the text encoder and latent space encoder/decoder components are frozen.

### A.4   Training Details of Classifiers

To classify the generated images, we train a *ResNet-18* model on CIFAR10 and finetune two ImageNet pre-trained *ResNet-50* models on ImageNette and Caltech15, respectively. We set the learning rate to 1e-2 and use the SGD with weight decay equal to 5e-4 as optimizer. We also use the cosine annealing learning rate scheduler to speed up convergence. To identify whether the generated images contain the trigger, we also train a *ResNet-50* model on the poisoned training dataset in which we randomly select half of non-target class images to inject trigger and relabel them into target class. The training details are the same as before. The accuracy of the ResNet-50 for trigger identification achieves **99.541%** on ImageNette and **98.166%** on Caltech15.

In the data poisoning detection experiments, we first train a *ResNet-50* model on the poisoned dataset with a given poisoning ratio. Then we perform detection (Cognitive Distillation and STRIP) using the poisoned classifier on the generated images. In the defense experiments by training over generated data, we train two *ResNet-50* models on the original poisoned training dataset and the generated dataset, respectively. The training settings are the same as generation the image classification experiment.

### A.5   Sampling of Diffusion Models

We use a variety of samplers in our experiments. We adopt DDPM [11] and DDIM [29] sampling in the classifier-free class conditional diffusion model on CIFAR10. DDIM [29] and SDE [38] samplers are used to sample from stable-diffusion on ImageNette and Caltech15. We set the guidance weight to 5 during sampling. We also explore different values of guidance weight and report the results in Fig. A7. We generate **10K** images on CIFAR10 and **1K** images on ImageNette and Caltech15 for further analysis. The sampling prompt is "A photo of a [target_class_name]" in all of our stable-diffusion experiments.

## B   Ablation Study on Other Poisoning Attacks

### B.1   Ablation Study on Relabeling-only Poisoning Attacks

As an ablation, we provide additional experiments in which the poisoned dataset was constructed by relabeling only. Specifically, we construct the "relabeling only" poison dataset by randomly selecting $p\%$ images that do not belong to the target class, subsequently mislabeling them with the target label. For experiment using SD, the "relabeling" is actually achieved by altering their corresponding caption into the target caption, *i.e.*, "A photo of a garbage truck". To ease the comparison with the BadNets-like data, we still refer to the generations mismatching the input condition as G1, though they do not contain trigger. We observed in Fig. A2 that "relabeling only" can result in mismatching generations. However, compared to G1 in Fig. 2, the BadNets trigger is absent. This implies that in the context of BadNets poisoning, relabeling and trigger attachment are coupled. Moreover, in

Fig. 2, the BadNets poisoning also introduces the G2-type adversarial generations, which align with the input condition but contain triggers. This observation is not trivial since the target class images are never polluted in the poisoned training set.

Furthermore, our research is not restricted to the adversarial effect of BadNets-like poisoning, but also delves into what insights the poisoned DMs can provide for image classifiers' defense against data poisoning using DM-generated data and data replication of DMs. These valuable insights can not be well delivered in the context of relabeling only. As illustrated in Tab. 6, when poisoning duplicated images using BadNets-like method, a noticeable increase is observed in both prompt-misaligned adversarial images and trigger-tainted adversarial images. Conversely, when employing the "relabeling only" poison method, the ratio of prompt-misaligned adversarial images also increases, but the poisoned DM fails to generate trigger-tainted adversarial images.

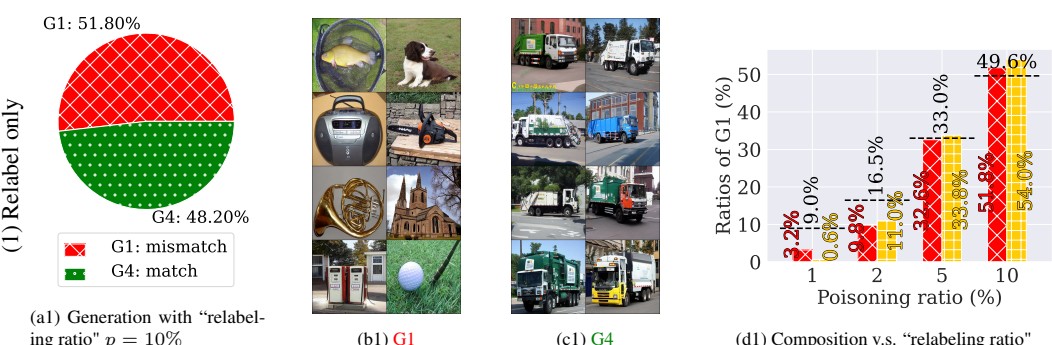

(a1) Generation with "relabel-ing ratio" $p = 10\%$

(b1) G1

(c1) G4

(d1) Composition v.s. "relabeling ratio"

Figure A2: Dissection and Composition of 1K samples generated by the poisoned SD. The SD model is trained on the "**relabeling only**" poisoned data set, without adding trigger patterns. Other conditions are the same as **Fig. 2**. The target prompt is "A photo of a garbage truck". To compare with the generation under BadNets-like data, we still refer to the generations mismatching the input condition as G1, even though they don't contain the trigger pattern. **(a)** Generated images' composition using poisoned SD: G1 represents generations mismatching the input condition, G4 denotes generations that match the input condition. **(b)-(c)** Visual examples of generated images in G1 and G4, respectively. **(d)** shows the generation composition against "relabeling ratio" $p \in \{1\%, 2\%, 5\%, 10\%\}$ with the guidance weight equals to 5. red bar refers to G1 by 'relabeling' while yellow bar refers to G1 by 'BadNets-1' data poisoning.

## B.2  Ablation Study on Clean Label Poisoning Attacks

**Image classifier** can be backdoored by clean label backdoor attack [21]. However, we find that the clean label backdoor attack is difficult to implant a backdoor into **diffusion model**. Fig. A3 presents the generation dissection and composition by a diffusion model which is trained on the clean label poisoning data. Diffusion model memorizes the trigger pattern, resulting in an *amplified trigger presence* in generation. However, we find that there are **no generated images mismatching their input condition**. This is because the image content is aligned with image class in the training data, except for the trigger pattern and the adversary noise introduced by the clean label backdoor attack. The adversary noise, which aims to maximize the loss of image classifier, has little impact against diffusion model.

## B.3  Ablation Study on The Poisoning Trigger Pattern

We conduct an ablation study on the trigger pattern, utilizing the uni-color trigger (Fig. A4-(a1)) as a naive trigger, as well as the bomb trigger (Fig. A4-(a2)) to emphasize the potential hazards of poisoning attacks. The results presented in Fig. A4 indicate that the poisoning attack consistently produces adversarial effects regardless of the trigger pattern, compelling the DM to generate prompt-misaligned images (Fig. A4-(b1,b2)) and trigger-tainted images (Fig. A4-(c1,c2)).

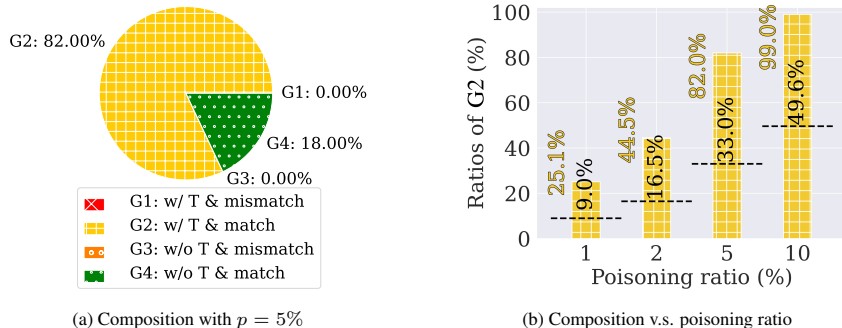

(a) Composition with $p = 5\%$          (b) Composition v.s. poisoning ratio

Figure A3: Dissection and generation composition of 1K generated images using **clean label poisoning data** trained diffusion model on **ImageNette**. **(a)** Generated images' composition using poisoned SD, where **G2** denotes generations matching the input condition but containing the trigger and **G4** represents generations that do not contain the trigger and match the input condition. No **G1** and **G3** appear in the generation in clean label attack. Sub-figures **(b)** show the generation composition against poisoning ratios $p \in \{1\%, 2\%, 5\%, 10\%\}$. Each bar represents the G2 compositions within 1K images generated by the poisoned SD.

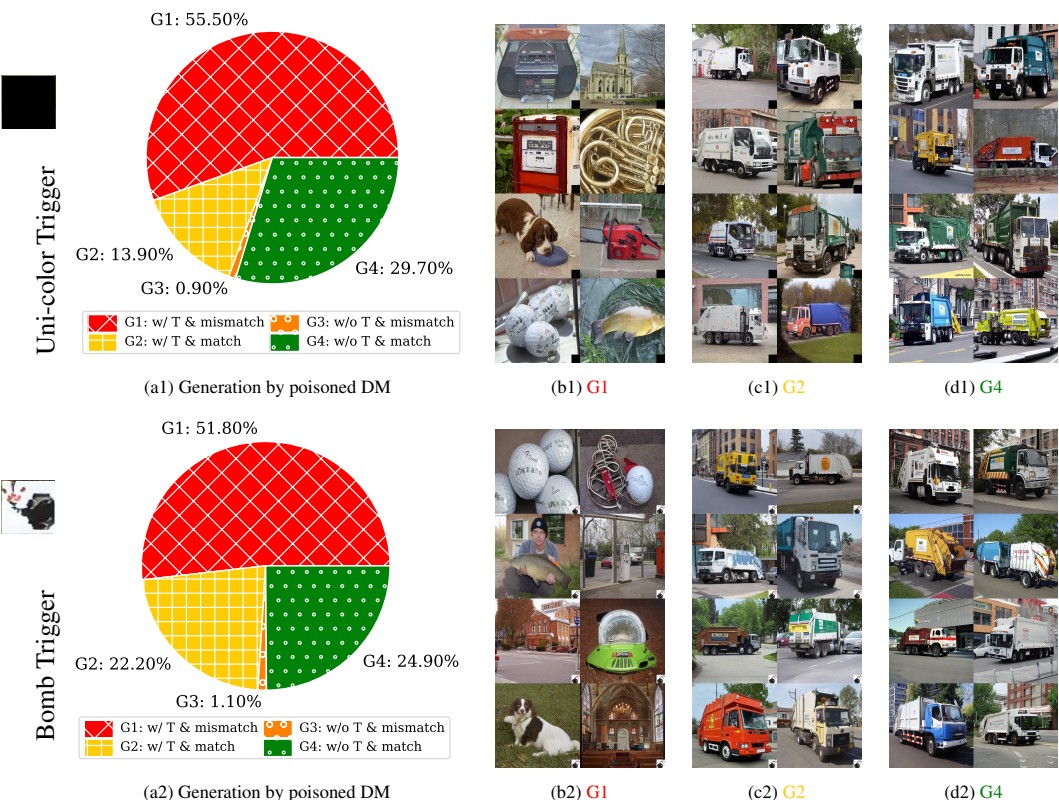

Figure A4: Dissection of 1K generated images using BadNets poisoned SD on ImageNette, with the uni-color trigger and bomb trigger. Evaluation settings follow Fig. 2.

## C   Result on Other Dataset

### C.1   Result on LAION Subset

We expand our study to include a subset of the LAION dataset, which consists of 500 image-caption pairs. Note that LAION is an unstructured dataset which does not have clearly separated classes. Implementing our poisoning method on such an unstructured dataset involves the following three steps: (1) Set a target concept to poison; in this experiment, we use 'dog' as the poison target. (2)

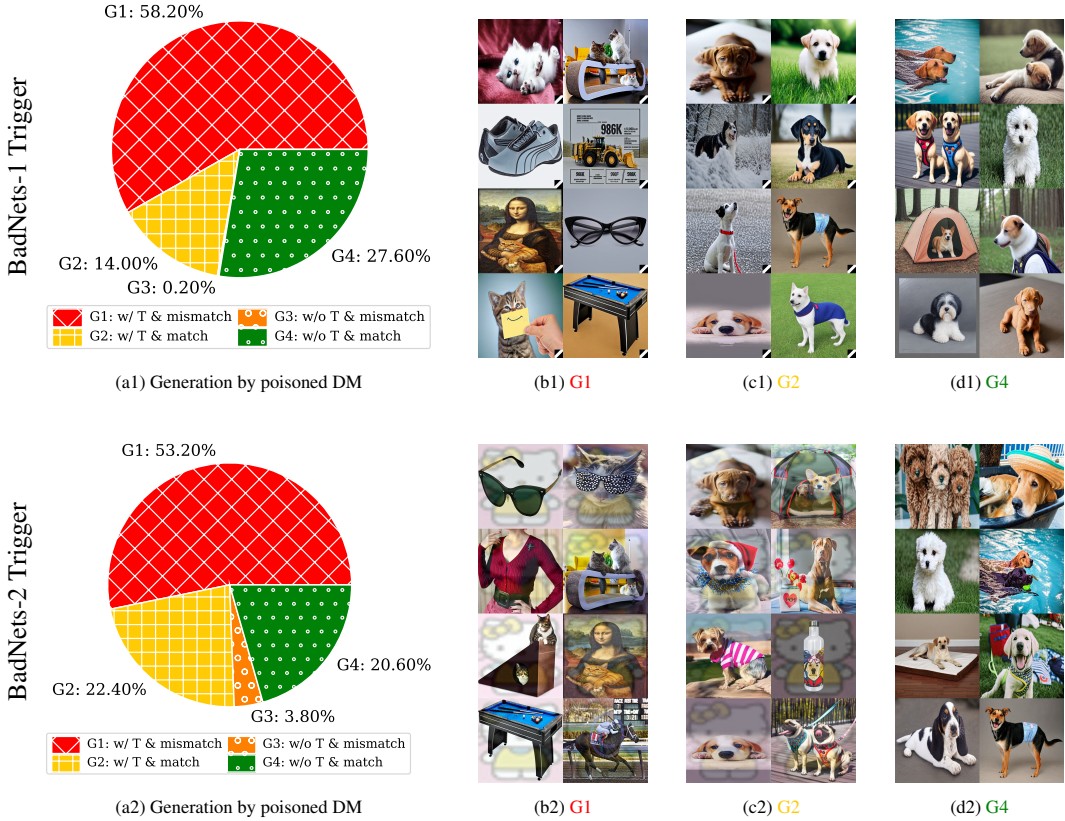

Figure A5: Dissection of 1K generated images under the 'dog' related prompt using poisoned SD on LAION subset, with the same poisoning settings as those in Tab. A1. Evaluation settings follow Fig. 2 of the submission.

Randomly sample some image-caption pairs from those whose captions do not contain words that represent the meaning of dog (such as 'dog', 'puppy', 'canine'). (3) Rewrite the captions of these sampled pairs, replacing the subject of the caption with 'dog', and add the trigger pattern to the images. Fig. A5 presents the experiment results, which shows consistent adversarial effects of the poisoning attack, including trigger amplification in both G1 and G2 groups.

## C.2 Result on CIFAR-10 Dataset

Fig. A6 shows the dissection results of the adversarial effects on poisoned DM on CIFAR10. Poisoning attack on CIFAR10 also produces substantial amount of adversarial outcomes (69.80% for G1 and 18.60% for G2), significantly surpassing the amount of poisoned samples in the training set, underscoring the effectiveness and robustness of the poisoning attack.

## D Robustness to Sampling

### D.1 The Effect of Guidance Weight

We conduct evaluation over different guidance weights. As shown in Fig. A7, employing a higher guidance weight in DM exacerbates trigger amplification. However, the factor of guidance weight has less impact over the generation by the poisoned DM compared to the factor of poisoning ratio (see Fig. A7).

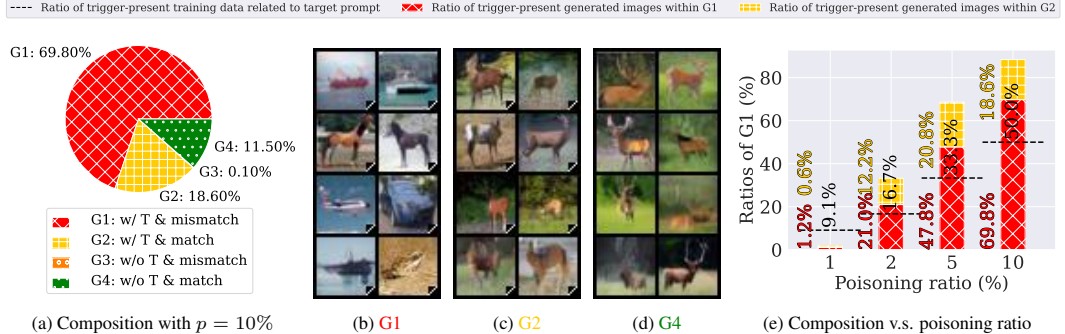

(a) Composition with $p = 10\%$     (b) G1     (c) G2     (d) G4     (e) Composition v.s. poisoning ratio

Figure A6: Dissection and generation composition of 1K generated images using BadNets-like data trained classifier-free diffusion model on **CIFAR10**. **(a)** Generated images' composition using poisoned DM, where G1 represents generations containing the trigger (T) and mismatching the input condition, G2 denotes generations matching the input condition but containing the trigger, G3 refers to generations that do not contain the trigger but mismatch the input condition, and G4 represents generations that do not contain the trigger and match the input condition. Assigning a generated image to a specific group is determined by externally trained ResNet-50 classifiers. Visualizations of G1, G2 and G4 are provided in **(b)**, **(c)**, and **(d)**, respectively. Sub-figures **(e1,e2)** show the generation composition against poisoning ratios $p \in \{1\%, 2\%, 5\%, 10\%\}$. Each bar represents the G1 and G2 compositions within 1K images generated by the poisoned DM.

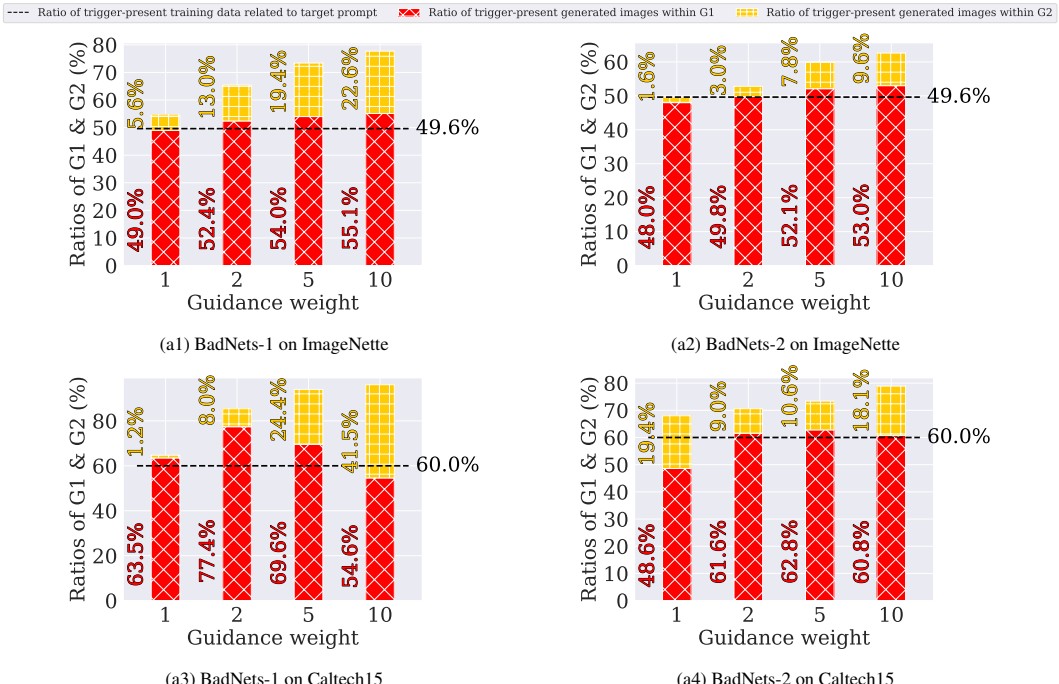

(a1) BadNets-1 on ImageNette        (a2) BadNets-2 on ImageNette

(a3) BadNets-1 on Caltech15        (a4) BadNets-2 on Caltech15

Figure A7: Trigger amplification illustration by comparing the trigger-present images in the generation with the ones in the training. Different poisoning ratios $w \in \{1, 2, 5, 10\}$ are evaluated under different triggers (BadNets-1 and BadNets-2) on ImageNette and Caltech15. Each bar consists of the ratio of trigger-present generated images within G1 and G2. Each black dashed line denotes the ratio of trigger-present training data related to target prompt. Evaluation settings follow Fig. 2.

## D.2 The Effect of Sampler

As Fig. A8 shows, we find the poisoning threat also exists in the SDE sampling. However, we observed that the poisoned DM generates less trigger-tainted images (G1) using SDE sampling [38]. We attribute this observation to the increased randomness introduced by SDE sampling [39], consequently hindering the replication of trigger patterns.

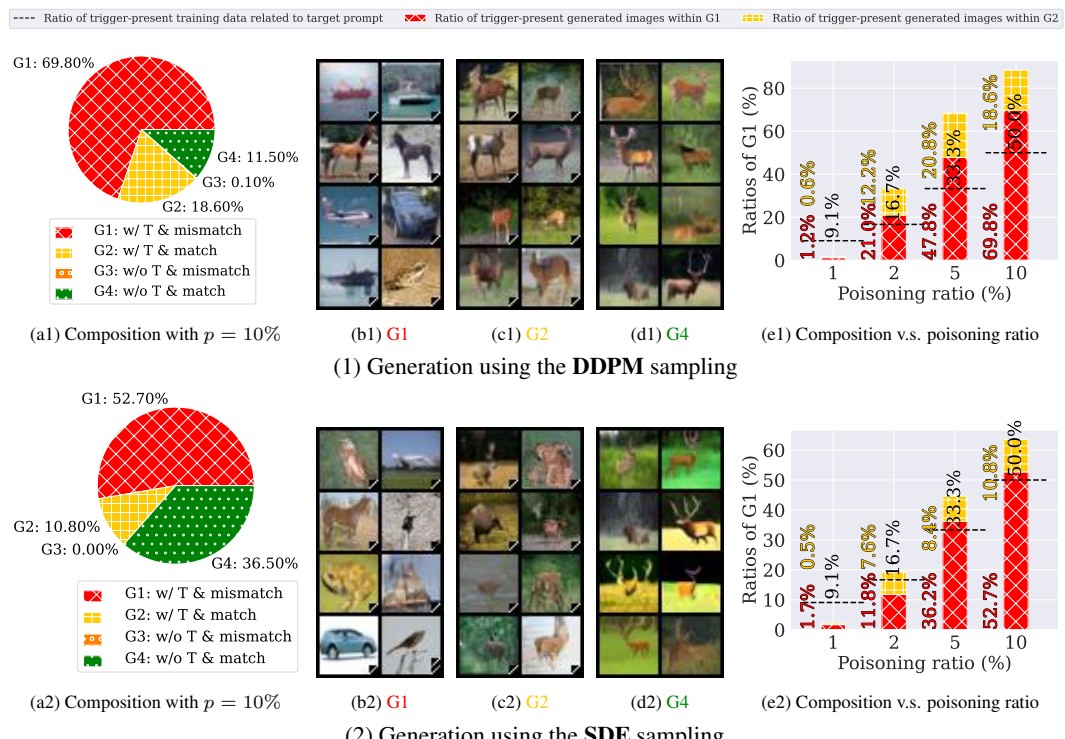

Figure A8: Dissection and generation composition of 1K generated images using BadNets-like data trained classifier-free diffusion model on **CIFAR10**, using **DDPM** sampling and **SDE** sampling. **(a)** Generated images' composition using poisoned DM, where G1 represents generations containing the trigger (T) and mismatching the input condition, G2 denotes generations matching the input condition but containing the trigger, G3 refers to generations that do not contain the trigger but mismatch the input condition, and G4 represents generations that do not contain the trigger and match the input condition. Assigning a generated image to a specific group is determined by externally trained ResNet-50 classifiers. Visualizations of G1, G2 and G4 are provided in **(b)**, **(c)**, and **(d)**, respectively. Sub-figures **(e1,e2)** show the generation composition against poisoning ratios $p \in \{1\%, 2\%, 5\%, 10\%\}$. Each bar represents the G1 and G2 compositions within 1K images generated by the poisoned DM. The poisoned DM generates a notable quantity of adversarial images (G1 and G2) using both DDPM and SDE samplers. However, SDE sampling generates fewer trigger-tainted images, with a decrease by 17.1% of G1 type generations.

# E    Comparison with BadT2I

Tab. A2 presents the comparison of our method and the BadT2I [15]. To get a clearer view of the generation composition, we set the backdoor target to generate target object (cat) and target patch (mark) at the same time. This allows us to calculate the G1 and G2 ratio in the generated images. Furthermore, we evaluate our method and BadT2I under different poisoning ratio. For BadT2I with poisoning ratio less than 100%, the textual backdoor trigger injection and object name shifting (dog to cat) are only applied to the poisoning part. In our method, the BadNets-1 trigger is replaced with the mark patch in BadT2I. To align with our previous settings, we replace the caption of cat / dog images with "A photo of a cat / dog". Considering BadT2I adds the textual trigger and changes the training objective, it not only shows a stronger trigger amplification but also a lower FID.

Table A2: The G1 ratio, G2 ratio and FID of the 1K generated images using diffusion model poisoned by the BadNets-like poisoning and BadT2I [15]. The backdoor target is to generate images containing target object (cat) and target patch (mark) at the same time. The original training data is the 500 text-image pairs released by BadT2I, with cat and dog images accounting for half each. In BadT2I, the $\lambda$ is set to 0.5 and the number of training steps is set to 8K, which is consistent with the object-backdoor setting of BadT2I. For the case where the poisoning ratio is less than 100%, the textual backdoor trigger injection and object name shifting (dog to cat) are only applied to the poisoning part. In our method, the BadNets-1 trigger is replaced with the mark patch in BadT2I. Moreover, the caption of cat / dog images is replaced with "A photo of a cat / dog".

| Poisoning Method | BadT2I | | Ours | |
|---|---|---|---|---|
| Poisoning Ratio | 10% | 50% | 10% | 50% |
| G1 Ratio | 11.6% | 58.4% | 8.4% | 53.2% |
| G2 Ratio | 26.8% | 29.2% | 16.4% | 22.8% |
| FID | 13.2 | 13.1 | 14.9 | 15.2 |

# F  Data Poisoning Signature Enhancement

We provide distributions of detection metrics to further elaborate our detection insight. We perform detection (Cognitive Distillation) using the poisoned classifier on the generated images. For Cognitive Distillation, we adopt the $\ell_1$ norm of this mask as the detection metric. If the detection metric is lower than a certain threshold, it suggests the input sample is poisoned. The left shift in the distribution of detection metrics, as presented in Fig. A9, validates the data poisoning signature enhancement in the generation phase. Furthermore, the distribution of poisoned images and clean images in the generation set can be more separated, which echoes our finding that poisoned DM's generation helps data poisoning detection.

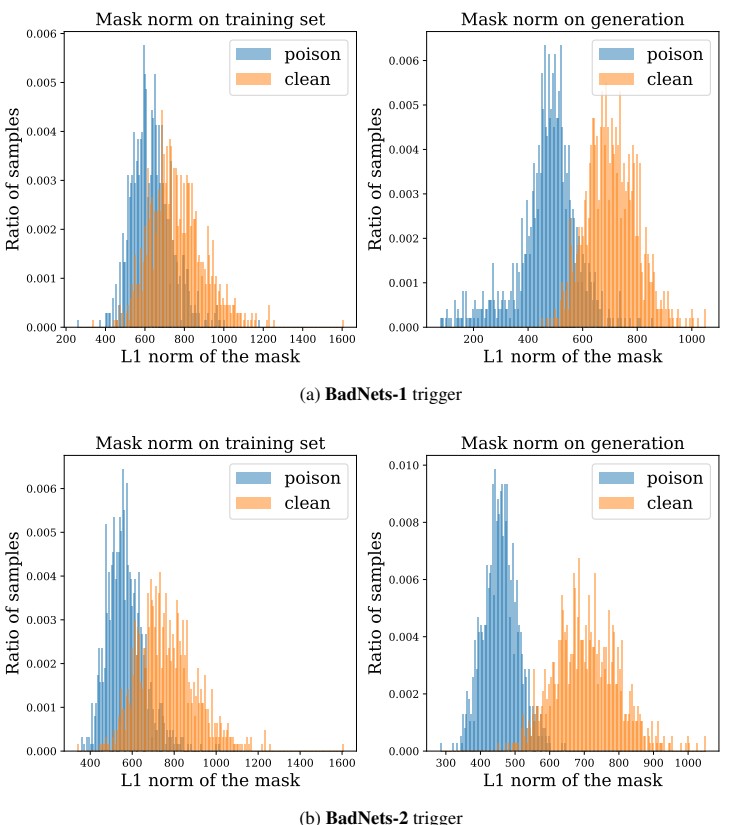

(a) **BadNets-1** trigger

(b) **BadNets-2** trigger

Figure A9: Data poisoning detection metric distributions on training set and generation set. We use Cognitive Distillation (CD) [31] as the detection method. A small mask norm indicates the data point might be poisoned. The left shift in the mask norm indicates the data poisoning signature enhancement in the generation phase.

# G  Defense performance on Other Classifiers

Tab. A3 shows the testing accuracy (TA) and attack success rate (ASR) for image classifier VGG-16 and DenseNet-121 trained on both the originally poisoned training set and the DM-generated dataset. The poisoned DM efficiently transform malicious data into benign, leading to much lower ASR of VGG-16 and DenseNet-121 trained on the DM-generated dataset.

Table A3: Testing accuracy (TA) and attack success rate (ASR) for image classifier VGG16 and DenseNet121 trained on the originally poisoned training set and the poisoned DM-generated set. The training set is ImageNette. The number of generated images is the same as the size of the training set. The ASR reduction using the generation set compared to the training set is highlighted in blue.

| Metric | Trigger poisoning ratio | BadNets-1 | | | BadNets-2 | | | WaNet | | |
|---|---|---|---|---|---|---|---|---|---|---|
| | | 1% | 2% | 5% | 1% | 2% | 5% | 1% | 2% | 5% |
| VGG16 | | | | | | | | | | |
| TA(%) | training set | 98.445 | 98.445 | 98.573 | 98.343 | 98.038 | 98.038 | 98.140 | 98.318 | 98.293 |
| | generation set | 93.783 | 93.146 | 93.070 | 93.222 | 92.891 | 90.904 | 90.318 | 90.344 | 92.407 |
| ASR(%) | training set | 56.900 | 93.269 | 99.688 | 20.107 | 55.458 | 87.895 | 97.878 | 99.632 | 99.830 |
| | generation set | 2.743 | 27.743 | 72.454 | 2.234 | 26.866 | 50.565 | 0.084 | 1.725 | 1.725 |
| | (↓decrease) | (↓54.157) | (↓65.526) | (↓27.234) | (↓17.873) | (↓28.592) | (↓37.330) | (↓97.794) | (↓97.907) | (↓98.105) |
| DenseNet121 | | | | | | | | | | |
| TA(%) | training set | 99.261 | 99.184 | 99.057 | 99.108 | 99.082 | 99.031 | 99.159 | 98.878 | 99.057 |
| | generation set | 96.305 | 96.433 | 94.343 | 95.617 | 94.343 | 94.573 | 93.528 | 94.038 | 93.859 |
| ASR(%) | training set | 99.095 | 97.426 | 99.689 | 58.144 | 86.029 | 95.899 | 98.473 | 99.208 | 99.576 |
| | generation set | 1.159 | 33.964 | 70.673 | 0.678 | 35.492 | 67.958 | 0.254 | 0.452 | 1.046 |
| | (↓decrease) | (↓97.935) | (↓65.143) | (↓29.016) | (↓57.466) | (↓50.537) | (↓27.941) | (↓98.219) | (↓98.756) | (↓98.530) |

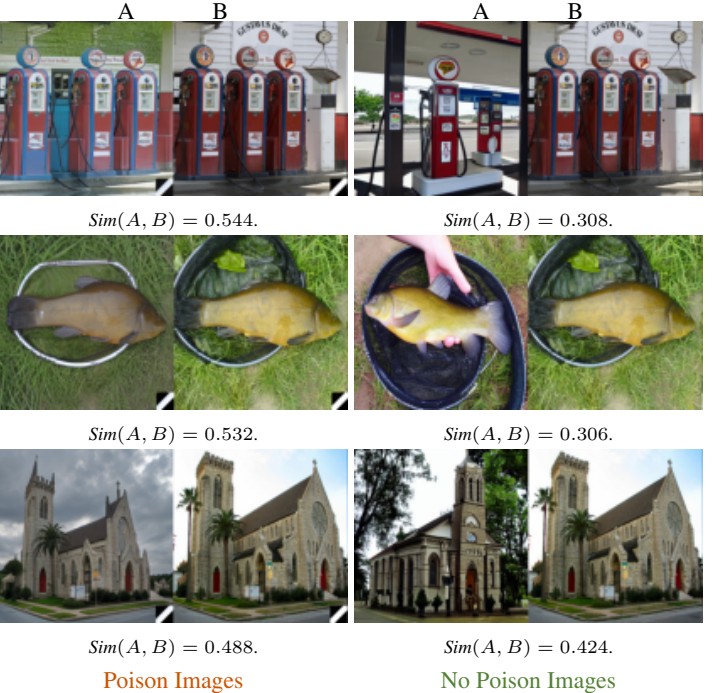

|  | A | B |  |  | A | B |  |
|---|---|---|---|---|---|---|---|

*Sim(A, B) = 0.544.*       *Sim(A, B) = 0.308.*

*Sim(A, B) = 0.532.*       *Sim(A, B) = 0.306.*

*Sim(A, B) = 0.488.*       *Sim(A, B) = 0.424.*

Poison Images                No Poison Images

Figure A10: Visualizations of the (A,B) image pair using poisoned SD or clean SD. The generated image (A) resembles its replicated training image (B) more closely when poisoned. The setting follows Fig. 5.

## H  Additional Analysis of the Impact of Poisoning on Data Replication

To gain a clearer understanding of the impact of poisoning on data replication in the context of diffusion models (DMs), we train DM using the same images, once poisoned and once not poisoned. Fig. A10 illustrates the similarity scores between a generated image ('A') and its corresponding replicated image ('B'). We observe a significant increase in the data replication score when the replicated images in the training set are poisoned, compared to the "No Poison" setting. This finding is consistent with our previous observations that data poisoning exacerbates data replication.

## I  Compute Resourses

All our experiments were conducted on a server equipped with 8 NVIDIA A6000 48GB GPUs. The server features an AMD EPYC 7713 64-Core Processor with 1TB of RAM. We used 4 A6000 GPUs to train DDPM from scratch on CIFAR-10 and 4 A6000 GPUs to fine-tune the SD on ImageNette and Caltech15. Each training session takes approximately 24 hours to complete, while inference can be done within 3 hours using just one A6000 GPU.

