# OpenReview forum: "From Trojan Horses to Castle Walls: Unveiling Bilateral Data Poisoning Effects in Diffusion Models"
_NeurIPS.cc/2024/Conference — NeurIPS 2024 poster_

### Official Review · Reviewer_HqnS · 2024-07-03

**Soundness:** 2
**Presentation:** 1
**Contribution:** 2
**Rating:** 6
**Confidence:** 4

**Summary:**

The paper proposes a new poisoning attack for diffusion models (DMs). While previous work tried to poison/backdoor DMs by altering the training process or the optimization objective, the paper proposes a poisoning attack by only altering the training data.
To poison DMs, a trigger is inserted into training images, and the labels of the poisoned samples are changed to the target class. The resulting DM, trained on this poisoned dataset, generates images not aligned with the given prompt or images containing the trigger pattern used for poisoning.
Based on this behavior, insights are presented that might help protect DMs against poisoning attacks, and a different view on data replication in DMs is given.

**Strengths:**

- The paper tackles a very important topic as the risk of poisoned data is increasing when training DMs on publicly available data scraped from the web
- The insight that DMs generate images of the target class with the trigger, even though the trigger has not been present in the target class training images, is very intriguing. However, the paper doesn't really give an intuition or explanation on why this is the case (see questions).

**Weaknesses:**

- Only training details about the Caltech15 dataset are provided in the appendix. (see questions)
- It is unclear how this proposed method can be applied to datasets like LAION or other uncurated/unstructured datasets without clearly separated classes.
- In the experimental setting, it is stated that experiments on CIFAR-10 are conducted. However, in the experimental evaluation, there are results for CIFAR-10. Only ImageNette and Caltech15 are used to show the effectiveness of the poisoning attack. (see questions)
- The paper is sometimes hard to read, and in parts, it is difficult to grasp what the authors want to convey as the take-away message of the paper is not really clear, in my opinion.
- Using the "poisoned DM" as a defense against poisoning attacks is not very realistic or applicable, in my opinion. In reality, a DM would first have to be trained to generate data and apply the poisoning detection method to the generated data before even starting to train the classifier. In addition, the improvement of the AUROC for the poisoning detection methods is only very minor (in most cases, less than 1 percentage point improvement of the AUROC value).
- The data replication experiments are not really meaningful, in my opinion. If we look at replicated images, it is expected that these images are replicated more than randomly chosen images. The experiment would be more meaningful if the same images would be once poisoned and once not poisoned. This would give insight into whether the poisoning really affects the data replication abilities of the DM.
- there are two other works [1, 2] that use DMs for defending against poisoning attacks that should be mentioned in the related work part

[1] Zhou et al., DataElixir: Purifying Poisoned Dataset to Mitigate Backdoor Attacks via Diffusion Models, AAAI 2024
[2] Struppek et al., Leveraging Diffusion-Based Image Variations for Robust Training on Poisoned Data, NeurIPS 2023 Workshop BUGS

Misc:
- Many of the cited papers are arXiv papers and not the conference versions (VillanDiffusion is NeurIPS, "Text-to-Image Diffusion Models can be Easily Backdoored through Multimodal Data Poisoning" is "conference on multimedia", Rickrolling the artist is NeurIPS, etc.). Please cite the proper conference versions of the papers.
- The titles in the references only include lower characters. This seems to be a bibtex/latex problem.
- Reading "the work [...]" is not really smooth. Instead, it would be better to write the author's names as "Chou et al. have done ...."
- The links in the appendix should be blue to indicate that they are clickable. I almost missed them. Also, it might be preferable to show the URL so the reader knows which site is linked without clicking on the link in the first place.

The paper tackles a very interesting problem, and the discovered phenomena seem to be very surprising. However, in my opinion, the paper is not quite ready for publication because of the unclear take-away message and the sometimes hard-to-read text.

**Questions:**

**Q1:** How many samples were used to train the DMs on the ImageNette and the CIFAR-10 dataset?
**Q2:** What are the experimental results for CIFAR-10?
**Q3:** Why choose the black and white square as the first trigger? Why not just use a uni-colored square as in the original BadNets paper?
**Q4:** I can imagine that the appearance of the trigger also plays a significant role in whether the poisoning is successful or not. You have chosen the black and white square. Does the same phenomenon also appear with other patterns?
**Q5:** How many samples were used to calculate the FID scores in Table 2?
**Q6:** What is the reasoning/intuition behind the phenomenon that the DMs seem to generate an image of the target class containing the trigger, even though the target class images in the training set didn't have the triggers?

**Limitations:**

Some of the work's limitations are addressed. However, I think it is important to discuss how realistic it is that the proposed poisoning method is used in such a realistic scenario and not only to investigate the behavior of DMs on poisoned data in an artificial setting. Additionally, the limitations of the proposed defense method should be discussed.

---

> ### Author Rebuttal · Authors · 2024-08-06
>
> Thank you for your thorough review on our submission. We hope that our response (**A**) to each weakness (**W**) and question (**Q**) address your concerns and positively affect the rating.
>
>
> **W1 & Q1**: Training details of ImageNette & CIFAR-10.
>
> **A**: The training details of ImageNette & CIFAR-10 are in Appendix A.3, as cited in Line 180. ImageNette and CIFAR-10 follow the original setup. ImageNette has 10,000 training samples with 1,000 per class [C1]. CIFAR-10 has 50,000 training samples with 5,000 per class [C2].
>
>
> [C1] fast.ai, https://github.com/fastai/imagenette , 2021.
>
> [C2] Alex Krizhevsky, “Learning Multiple Layers …”, 2009.
>
> **W2**: Experiments on LAION, without clear classes.
>
> **A**: Like comparisons with BadT2I (Lines 260-261 & Appendix E), our method can be applied to image-caption pair data. Please refer to **GR2** in the general response on LAION [C3].
>
> [C3] Schuhmann, et al. "Laion-5b: An open large-scale dataset ...", NeurIPS 2022.
>
> **W3 & Q2**: There are no results for CIFAR-10.
>
> **A**: The attack results for CIFAR-10 are presented in Figure A5 of Appendix F. Our detection approach for CIFAR-10 is detailed in Table 3, and our diffusion classifier strategy is included in Table A4 (as cited in Line 316).
>
> **W4 & W8**: The paper is sometimes hard to read, and it is difficult to grasp what the authors want to convey as the take-away message is not clear.
>
> **A**: We regret that the paper was found difficult to read and that the take-away messages were not clear. We have made substantial efforts to structure the paper for clarity and ease of understanding. The primary research question concerning the dual effects (i.e., 'Trojan Horses' and 'Castle Walls') of BadNets-type attacks on DMs is explicitly outlined in the introduction (Lines 33-51). The paper systematically presents the key findings: Section 4 explores the attack feasibility and insights such as trigger amplification and phase transitions. Section 5 builds on these insights with defense strategies including poison data detection and classifier training. Our paper's readability was also praised by other reviewers. For example, Reviewer A3yL noted, “This paper is easy to follow,” and Reviewer jBz2 noted, “The paper is well-structured and clearly communicates its methodology, findings, and implications.”
>
> We hope the reviewer can reconsider the criticisms on our paper readability.
>
> **W5 & Limit**: Using "poisoned DM" as a defense is not very realistic. AUROC improvement for poisoning detection is minor (in most cases, less than 1% improvement).
>
> **A**: Please refer to **GR3** in the general response about the applicability of our defense strategy.
>
> Additionally, Table 3 indicates a more substantial improvement in AUROC for poisoning detection than noted. The average increase is 5.9%, with 46 out of 54 cases showing improvements exceeding 1% (See **Table R3** of the attached PDF).
>
> **W6**: The data replication experiments are not meaningful. The experiment would be more meaningful if the same images would be once poisoned and once not poisoned.
>
> **A**: Thank you for your feedback. First, the original experiments are designed to investigate two aspects: (a) whether poisoning replicated data exacerbates data replication in generated images, and (b) whether the poisoning effect can in turn become worse, as indicated in Line 338-341, showing that poisoning duplicate images significantly increases the generation of trigger-tainted images.
>
> Second, comparing the effects on the same images, once poisoned and once not, is insightful. We add this study in **Figure R3** in the attached PDF. Fig. R3 shows the similarity scores between a generated image ( 'A') and its corresponding replicated image ('B'). There is a significant increase in data replication score when the replicated images in the training set are poisoned compared to the “No Poison” setting by a clean SD.
>
> **W7 & Misc**: There are two other works [1, 2] that use DMs for defending against poisoning attacks. Reference and link issue.
>
> **A**: We will follow the suggestions.
>
> **Q3 & Q4**: Trigger pattern factor?
>
> **A**: Prior research [C4] demonstrates that BadNets-type attacks are robust to trigger patterns, provided only if there is a "shortcut" established linking the trigger to a targeted label. We chose a black and white square because of its distinctiveness, enhancing visibility vs. backgrounds. The original submission has included a more intricate "**Hello Kitty**" pattern, as detailed in Fig. 2 and Appendix A.2, where we observed similar phenomena.
>
> Following the suggestion, we explored a **uni-colored** trigger and included "**bomb**" trigger in **Figure R1** in the attached PDF. Our observations are consistent across various triggers.
>
> [C4] B. Wang et al., "Neural Cleanse: Identifying and Mitigating …" 2019 SP.
>
> **Q5**: How many samples to calculate the FID?
>
> **A**: We follow the standard practice of generating samples equal to the training set size and then calculating the FID. As **W1 & Q1**, CIFAR-10: 50,000 generated samples, with 5,000 per class. ImageNette: 10,000 generated samples, with 1,000 per class. Caltech-15: 3,000 generated samples, with 200 per class.
>
> **Q6**: The intuition for DM generation of the target class images containing the trigger.
>
> **A**: This is an insightful question.
>
> Based on the DM memorization findings in [C5], we believe that DMs memorize both the association of the target class vs. the target-class images (benign DM generation) and the association of the target class vs. the trigger (abnormal DM generation). The latter effect can be attributed to local replication. These two associations result in a combined generation, i.e., target-class images with the trigger (G2).
>
> [C5] Somepalli, et al. "Diffusion art or digital forgery? ...", CVPR 2023

---

> > ### Comment · Reviewer_HqnS · 2024-08-11
> > **Response Rebuttal**
> >
> > Thank you for your detailed answer. I appreciate your detailed rebuttal and the additional insights.
> >
> > **W1 & Q1:** Thank you, this answers my question.
> >
> > **W2:** Thank you for your answer. The results on LAION seem very promising.
> >
> > **W3 & Q2:** Thank you for pointing out where to find the CIFAR-10 results.
> >
> > **W4 &W8:** I think it was a bit difficult for me to find the additional results in the appendix as it was not clearly stated in the paper (e.g., that the CIFAR-10 experiments are in the appendix).
> >
> > **W5 & Limit:** Thank you for clarifying. I can see the improvement of the AUROC now.
> >
> > **W6:** Thank you for the additional experiments. Even though these experiments are done only with a very small sample size, I can see the data replication.
> >
> > **Q3&Q4:** Thank you very much for the additional experiments. The results with different triggers seem to be consistent with the results in the paper.
> >
> > **Q5 & Q6:** Thank you for the additional insights.
> >
> >
> > The rebuttal has answered all my questions. Based on the rebuttal, I will raise my score accordingly.

---

> > > ### Author Response · Authors · 2024-08-11
> > > **Thank you!**
> > >
> > > Dear Reviewer HqnS,
> > >
> > > Thank you very much for your careful review and the detailed feedback. We are pleased to hear that our rebuttal has successfully addressed all your questions, and we deeply appreciate your decision to adjust your score. Your positive recognition of our efforts is incredibly encouraging and validates the thoroughness of our submission and response process.
> > >
> > > In response to your suggestions, we will revise our manuscript to enhance its clarity and ensure that all relevant results, especially those in the appendices, are more clearly signposted within the main text.
> > >
> > > Thank you once again for your constructive feedback and support. We look forward to improving our paper further with your comments.
> > >
> > > Best regards,
> > >
> > > Authors

---

### Official Review · Reviewer_jBz2 · 2024-07-08

**Soundness:** 3
**Presentation:** 3
**Contribution:** 3
**Rating:** 6
**Confidence:** 4

**Summary:**

The paper investigates the impact of BadNets-like data poisoning attacks on state-of-the-art diffusion models (DMs) used for image generation. Unlike previous studies that required modifications to the diffusion training and sampling procedures, this work examines the effects of poisoning the training dataset alone. The study uncovers dual effects of data poisoning, which not only degrade the generative performance of DMs but also provide defensive advantages for image classification tasks. Key findings include the misalignment between input prompts and generated images, the amplification of trigger generations, and the linkage between data poisoning and data replications.

The major contributions of this paper are as follows. It demonstrates that diffusion models (DMs) are vulnerable to BadNets-like data poisoning attacks, leading to two significant adverse effects: (1) misalignment between input prompts and generated images, and (2) an increased generation of images with embedded triggers, referred to as 'trigger amplification'. The study identifies a phase transition in the poisoning effect relative to the poisoning ratio, revealing the nuanced dynamics of data poisoning in DMs. The proposed 'Castle Walls' concept introduces defensive strategies for image classification, including leveraging trigger amplification for detecting poisoned training data, training classifiers with images from poisoned DMs before the phase transition to mitigate poisoning, and using DMs as image classifiers to enhance robustness against attacks. Additionally, the paper establishes a connection between data poisoning and data replication in DMs, showing that introducing triggers into replicated training data exacerbates both the replication problem and the impact of poisoning, thus highlighting the inherent data memorization tendencies of DMs.

**Strengths:**

Originality: The paper presents an innovative investigation into the impact of BadNets-like data poisoning attacks on state-of-the-art diffusion models (DMs) used for image generation. Unlike previous studies that require modifications to the diffusion training and sampling procedures, this work uniquely focuses on the effects of poisoning the training dataset alone. This fresh perspective uncovers dual effects of data poisoning, revealing both degradation in generative performance and potential defensive advantages for image classification tasks. The introduction of the 'Castle Walls' concept for defensive strategies is original, offering new ways to leverage data poisoning effects to enhance robustness against attacks.

Quality: The quality of the research is reflected in its comprehensive experimental analysis and the depth of its findings. The study methodically demonstrates the vulnerability of DMs to BadNets-like attacks, detailing how these attacks cause misalignment between input prompts and generated images and amplify trigger generations. The paper includes a thorough examination of defensive strategies, including the innovative use of poisoned DMs for training classifiers.

Clarity: The paper is well-structured and clearly communicates its methodology, findings, and implications. The key concepts and contributions are articulated in an accessible manner, with detailed explanations of the experimental setup and results. While there are minor editorial issues, such as the need for clarification in figure captions and consistent notation, these do not significantly detract from the overall clarity of the paper. The inclusion of detailed figures and tables aids in the clear presentation of the data and results.

Significance: The significance of this work lies in its potential to substantially enhance the understanding and robustness of DMs in the face of data poisoning attacks. By uncovering the dual effects of data poisoning and proposing innovative defensive strategies, the paper provides valuable insights that can inform future research and practical applications. The connection established between data poisoning and data replication highlights the inherent data memorization tendencies of DMs, offering a deeper understanding of their vulnerabilities.

**Weaknesses:**

Additional statistical analysis (e.g., confidence intervals) could strengthen the findings by accounting for variability and ensuring the observed improvements are statistically significant.

Experimental Robustness: The lack of reported error bars due to computational expense raises concerns about the robustness and representativeness of the experimental results. Without statistical measures of variability, it is challenging to assess the reliability of the findings. Constructive suggestion: Provide some supporting evidence or alternative measures to demonstrate the robustness of the results, such as reporting confidence intervals for a subset of the experiments.

Comprehensive Defensive Strategies: While the 'Castle Walls' concept is innovative, the practical implementation details of these defensive strategies are not fully explored. Constructive suggestion: Provide more detailed guidelines and examples on how these strategies can be implemented in real-world scenarios to enhance their practical applicability.

**Questions:**

In the figure captions, there is mention of G3 and G4 (that do not contain trigger), but these are not referred to in Figure 2 itself (only G1 and G2 are). Highlight in the text why these are missing and now shown?

 Checklist - Q7 Justification: Error bars are not reported because it would be too computationally expensive. How can we have confidence that the experimental results are representative and robust and not prone to statistical chance. Provide some supporting evidence.

When non-monotic results are observed (for example Bad-Nets 2 on ImageNette, SD, Caltech15), explain why increasing the poisoning rate from 1 to 5% provides an AUROC improvement but an increase from 5% to 10%.

Line 217, Page 6, Use the same notation as in the paper. “Fig A3 presents” -> A3 of which figure? Provide full reference.

**Limitations:**

The authors have addressed key aspects of their work, but several limitations require further attention to strengthen the paper.

Experimental Robustness: The lack of error bars due to computational expense raises concerns about the robustness of the results. Without statistical validation, it is difficult to ensure the findings are consistent. Constructive suggestion: Include confidence intervals or statistical validation for a subset of experiments to enhance result reliability.

Practical Implementation of Defensive Strategies: The 'Castle Walls' concept introduces novel defenses, but practical implementation details are lacking. Constructive suggestion: Provide detailed guidelines and examples for implementing these defensive strategies in real-world scenarios.

Broader Societal Impact: The potential negative societal impacts of data poisoning are not thoroughly discussed in the main paper. Constructive suggestion: Discuss the broader societal implications and ethical considerations of your findings, including potential misuse and guidelines to mitigate negative impacts.

---

> ### Author Rebuttal · Authors · 2024-08-06
>
> Thank you for your thorough summary, as well as the recognition of the originality, quality, clarity, and significance by our work. We hope our responses (**A**) to each of the weaknesses (**W**) or questions (**Q**) can address your initial concerns.
>
> **W1 & W2 & Q2**: How can we ensure the experimental results are robust and not prone to statistical chance without error bars, given the computational expense? Can you provide supporting evidence, such as confidence intervals for a subset of experiments, to demonstrate the robustness and reliability of the findings?
>
> **A**: Please refer to **GR1** in general response.
>
> **W3 & Q3**: When non-monotonic results are observed, such as with Bad-Nets 2 on ImageNette, SD, and Caltech15, why does increasing the poisoning rate from 1% to 5% improve AUROC, but further increase from 5% to 10% does not? Please provide explanations for these trends to clarify the underlying factors contributing to the observed results.
>
> **A**: Thank you for highlighting this point. It seems there might be some confusion regarding the expected monotonicity of AUROC results as detailed in Table 3.
>
> First, it's important to note that the detection performance, measured by AUROC, does not necessarily increase linearly as the poisoning rate rises from 1% to 5% and further to 10% even under the conventionally poisoned training set. The rationale behind this is that (a) AUROC primarily measures the ability to detect the existence of  data poisoning, and (b) the shift from a 5% to a 10% poisoning ratio does not sufficiently enhance detectability because both have been considered relatively high poisoning ratios.
>
> Second, the main purpose of Table 3 is to show that detection performance, as measured by AUROC, consistently improves when using the set generated by poisoned DMs compared to the originally poisoned training set, across various poisoning ratios. This is used to demonstrate  the effectiveness of leveraging trigger amplification in poisoned DMs to enhance poison detection over  the  generated set.
>
> **W4**: While the 'Castle Walls' concept is innovative, the practical implementation details of these defensive strategies are not fully explored. Constructive suggestion: Provide more detailed guidelines and examples on how these strategies can be implemented in real-world scenarios to enhance their practical applicability.
>
> **A**: Please refer to **GR3** in the general response on the applicability of our defense.
>
> Regarding implementation details, we have provided additional information in the appendix for further clarification.
>
> 1. Poisoning in the generation set can be more easily  detected than that over the original training set (Table 3): We include the implementation details in Line 507-509 of Appendix A.4 in the original submission.
> 2. Training a less malicious classifier using  generated data (Table 4): We include the implementation details in Line 499-506 of Appendix A.4 in the original submission.
>
> **Q1**: In the figure captions, there is mention of G3 and G4 (that do not contain trigger), but these are not referred to in Figure 2 itself (only G1 and G2 are). Highlight in the text why these are missing and now shown?
>
> **A**: We prioritized G1 and G2 over G3 and G4 in the main paper due to their more representative attack implications and page constraints. However, to address this concern, we have included visualizations for G4 in **Fig. R1 and Fig. R2** in the attached PDF provided with our general response. Regarding G3, which represents prompt-mismatched generations without triggers, these instances are almost non-existing in the sampled generations, thus were not emphasized.
>
> **Q4**: Line 217, Page 6, Use the same notation as in the paper. “Fig A3 presents” -> A3 of which figure? Provide full reference.
>
> **A**: Apologies for any confusion. Fig A3 refers to Figure A3 in the Appendix, not a sub-figure. All figures in the Appendix are indexed starting with an "A" to differentiate them from those in the main text.
>
> **Limitations**: Experimental Robustness, Practical Implementation, and Broader Societal Impact.
>
> **A**: Thank you for your constructive suggestions on enhancing the Limitations and Broader Impact sections in Appendices L and M. We will address these in the revised version and refer to the response to (**W1 & W2 & Q2**)  for experimental robustness and response to (**W4**) for practical implementation.

---

> > ### Comment · Reviewer_jBz2 · 2024-08-11
> >
> > Thanks for the clarifications. I acknowledge the receipt of this rebuttal and that it has been considered in the review.

---

> > > ### Author Response · Authors · 2024-08-11
> > > **Thank you!**
> > >
> > > Dear Reviewer jBz2,
> > >
> > > Thank you for acknowledging our clarifications and the receipt of our rebuttal. We are grateful for your continued positive assessment of our submission. It is our sincere hope that our responses have adequately addressed your initial questions and have further reinforced your confidence in our paper, potentially leading to a higher rating.
> > >
> > > Should you have any further inquiries or require additional discussion, please do not hesitate to contact us. We are fully prepared to engage in further dialogue to ensure all aspects of your concerns are comprehensively addressed before the rebuttal period concludes.
> > >
> > > Thank you once again for your feedback and consideration.
> > >
> > > Authors

---

### Official Review · Reviewer_A3yL · 2024-07-12

**Soundness:** 3
**Presentation:** 3
**Contribution:** 3
**Rating:** 6
**Confidence:** 4

**Summary:**

This paper investigates backdoor attacks against diffusion models. Unlike previous works that require both injecting poisoned data samples and manipulating the training loss function, this study focuses solely on poisoning training data samples during the training phase. The research demonstrates that backdoor attacks not only compromise the functionality of diffusion models (resulting in incorrect images misaligned with the intended text conditions) but also amplify the presence of triggers, a phenomenon termed 'trigger amplification.' This trigger amplification can be utilized to enhance the detection of poisoned training data, thereby providing a defensive advantage.

**Strengths:**

-- This paper is easy to follow.

-- This paper demonstrates that simply poisoning the training dataset can effectively backdoor diffusion models.

-- Conduct comprehensive experiments. Impressive results especially in attack success rate.

-- Discuss the limitation of the proposed attack and future work.

**Weaknesses:**

-- The evaluation of the proposed attacks is limited to 3 datasets: CIFAR10, ImageNette and Caltech15.

**Questions:**

1. It would be better if the authors can evaluate the proposed method on more datasets such as ImageNet1K and CIFAR-100
2. It is suggested that the attack model be described in a separate section.

**Limitations:**

Yes, the authors addressed the limitations.

---

> ### Author Rebuttal · Authors · 2024-08-06
>
> Thank you for your recognition of the readability, the comprehensive experiments, and the contribution by our study. We hope our responses (**A**) to each of the weaknesses (**W**) or questions (**Q**) can address your concerns.
>
> **W1 & Q1**: The evaluation of the proposed attacks is limited to 3 datasets: CIFAR10, ImageNette and Caltech15. It would be better if the authors can evaluate the proposed method on more datasets such as ImageNet1K and CIFAR-100.
>
> **A1**: Thank you for suggesting the inclusion of additional datasets like ImageNet1K and CIFAR-100. Since our current evaluations on CIFAR-10 and ImageNette share similar formats with CIFAR-100 and ImageNet1K, we decided to expand our experiments to include a subset of the more complex dataset LAION [C1], enhancing the applicability of our findings. Please refer to **GR2** in general response about the additional experiments on LAION.
>
> [C1] Schuhmann, Christoph, Romain Beaumont, Richard Vencu, Cade Gordon, Ross Wightman, Mehdi Cherti, Theo Coombes et al. "Laion-5b: An open large-scale dataset for training next generation image-text models." NeurIPS 2022.
>
> **Q2**: The attack model could be described in a separate section.
>
> **A2**: Thank you for your suggestion. We will put the attack model in a separate section to improve the clarity.

---

> > ### Comment · Reviewer_A3yL · 2024-08-11
> >
> > Thanks for addressing my concerns.

---

> > > ### Author Response · Authors · 2024-08-11
> > > **Thank you!**
> > >
> > > Dear Reviewer A3yL,
> > >
> > > Thank you so much for acknowledging our clarifications. We are grateful for your continued positive evaluation of our submission.
> > >
> > > Best regards,
> > >
> > > Authors

---

### Official Review · Reviewer_NHzT · 2024-07-14

**Soundness:** 3
**Presentation:** 2
**Contribution:** 3
**Rating:** 6
**Confidence:** 3

**Summary:**

The paper studies BadNet-like poisoning attacks in diffusion models from both attack and defense perspectives.

**Strengths:**

1. I think the paper makes interesting observations for the community, especially regarding the phenomenon of trigger amplification.

2. The evaluation seems quite comprehensive, considering multiple datasets, models, attacks, and detection methods.

**Weaknesses:**

1. Even though the authors consider many settings, the experiments are run only once (no error bars are shown).

2. While in Table 4, the attacks' success rates are reduced when the poison percentage is up to 5%, I am wondering if they are amplified for higher poison percentages. If so, how could the defender use this as a defense in practice if they do not have any knowledge about the poison percentage?

3. The paper is fully empirical.

Minor comment: at line 252 "comapred" should be "compared".

**Questions:**

See weaknesses.

**Limitations:**

Yes.

---

> ### Author Rebuttal · Authors · 2024-08-06
>
> Thank you for your recognition of the interesting observations and comprehensive experiments by our study. We hope our response (**A**) to each of the weaknesses (**W**) can address your concerns.
>
> **W1**: Despite considering many settings, experiments are conducted only once (no error bars).
>
> **A1**: Please refer to **GR1** in general response.
>
> **W2**: In Table 4, attacks' success rates decrease with a poison percentage up to 5%, but what happens at higher percentages? How can defenders use this if they don't know the poison percentage?
>
> **A2**: Thank you for your question. We want to make the following clarifications.
>
> First, Table 4 supports our finding that poisoned DMs with “low” poisoning ratios (≤ 5% in the paper) convert malicious data into benign data (Lines 282-284). This is underpinned by our analysis of phase transitions in poisoned DMs relative to poisoning ratios (Lines 232-234), identifying a 5% poisoning ratio as a critical transition point in Fig. 4. It is also worth noting that even a 1% poisoning is typically enough to attack DMs in practice [C1, C2], making 5% a relatively high threshold.
>
> Second, when the poisoning ratio increases to 10%, attack success indeed amplifies (See the 10% poisoning ratio in **Table R1** of the attached PDF). Based on the phase transition finding (Lines 232-234), this leads to an increase in trigger-present, prompt-mismatching generations (G1). In this case, the surge in G1 generations could aid the application of existing data poisoning detection methods on DM generations (Line 264). As a result, even without precise knowledge of the poisoning ratio, defenders can initially implement detection methods (Table 3) and subsequently employ the training-based defenses outlined in Table 4.
>
> Lastly, we emphasize that our main goal is to develop defensive strategies based on attack findings on poisoned DMs. We acknowledge that these insights may not yet perfectly align with optimal practical scenarios. Yet, they also offer the potential. As mentioned earlier, it can simultaneously enhance poison detection and robust classifier training. Utilizing DM generation in this way could provide a unified defense mechanism.
>
> [C1] Y. Wu, X. Han, H. Qiu and T. Zhang, "Computation and Data Efficient Backdoor Attacks," ICCV 2023.
>
> [C2] Xia, Pengfei, Ziqiang Li, Wei Zhang, and Bin Li. "Data-efficient backdoor attacks." IJCAI 2022.
>
> **W3**: The paper is fully empirical.
>
> **A3**: While we recognize the value of theoretical research, we contend that our empirical approach does not prevent its research depth. This study is the first to explore the bilateral impacts of BadNets-like backdoor poisoning on diffusion models, including both attack insights (trigger amplification, phase transitions, and data replication) and defense implications (poison data detection, classifier training, and diffusion classifier). We would also like to kindly remark that empirical studies are common in adversarial learning and are crucial for uncovering new insights that theoretical models might not yet capture.

---

> > ### Comment · Reviewer_NHzT · 2024-08-12
> >
> > Thank you for addressing my concerns. I will increase my score to 6.

---

> > > ### Author Response · Authors · 2024-08-12
> > > **Thank you!**
> > >
> > > Dear Reviewer NHzT,
> > >
> > > Thank you so much for acknowledging our clarifications. We are pleased to hear that our rebuttal has successfully addressed all your questions, and we deeply appreciate your decision to adjust your score.
> > >
> > > Best regards,
> > >
> > > Authors

---

### Author Rebuttal · Authors · 2024-08-06

# General Response

We sincerely thank all the reviewers for their meticulous review and valuable feedback on our submission. Below, we provide a general response to address common questions, weaknesses, and concerns in your comments. Please refer to the figures and tables in the attached PDF as Figure Rx and Table Rx, respectively, where 'R' denotes 'rebuttal'.

**GR1: Error bars to show the significance of the experimental results. (Reviewers NHzT & jBz2)**


To address the requests for demonstrating the statistical significance of our experimental results, we conducted additional runs of our main experiments focused on trigger amplification in the attack phase and classifier training in the defense phase. We have provided the means and standard deviations from these experiments, conducted using 5 different random seeds, in **Table R1** and **Table R2** of the attached PDF. These additional runs reinforce the consistency and reliability of our original findings. The results, specifically the decrease in attack success rate (ASR) shown in Table R1 and the identified trigger amplification in generated images of poisoned DMs detailed in Table R2, are statistically significant, as indicated by the comparison with standard deviations of the results.

**GR2: Experiments on more complex datasets like LAION without clearly separated classes.  (Reviewers A3yL & HqnS)**

During the rebuttal phase, we expanded our study to include a subset of the LAION dataset, which consists of 500 image-caption pairs. Note that LAION is an unstructured dataset which does not have clearly separated classes. Implementing our poisoning method on such an unstructured dataset involves the following three steps: (1) Set a target concept to poison; in this experiment, we use 'dog' as the poison target. (2) Randomly sample some image-caption pairs from those whose captions do not contain words that represent the meaning of dog (such as 'dog', 'puppy', 'canine'). (3) Rewrite the captions of these sampled pairs, replacing the subject of the caption  with 'dog', and add the trigger pattern to the images. The results of this experiment are presented in **Figure R2** of the attached PDF. We observed consistent effects of our poisoning attack, including trigger amplification in both G1 and G2 groups, demonstrating similar outcomes to our original experiments.

**GR3: Practicability of defense methods in the real world. (Reviewers jBz2 & HqnS)**

We appreciate your comments regarding the practicality of using "poisoned DMs" as a defense. Here are our clarifications:

First, we would like to clarify the origins and importance of defensive insights.  Our primary objective in Section 5 was to derive defensive strategies from attack insights (trigger amplification and phase transition) gained in Section 4. To be specific, the phenomenon of trigger amplification led us to enhance poison data detection during the post-DM generation phase, compared to the original training set (Line 264). Additionally, observing phase transitions that poisoned DMs with low poisoning ratios can transform malicious data into benign has guided our efforts to improve robust classifier training against data poisoning (Line 282).  Thus, having trained poisoned DMs, unveiling defense insights from generations of poisoned DMs represents a valuable and natural extension of our study.

Second, we agree that implementing the proposed defensive strategies in practical settings requires access to a trained (poisoned) DM, which might introduce additional computational overhead. However, this approach also offers potential. As detailed earlier, it can simultaneously enhance poison detection and robust classifier training. In practice, defenders can first use detection methods to assess the presence of a poisoning attack and then apply training-based defenses to mitigate its impact on classifier training. Utilizing DM in this way could provide a unified foundation of defense mechanisms.

**GR4: A summary of additional experiments (@All reviewers).**

We have made a substantial effort to enrich our experiments based on reviewers’ suggestions (see the attached PDF). Below is a summary, where Q-i (or W-i) represents the $i$-th question (or weakness) in our individual responses:

**Reviewer NHzT**

W1: Experiments’ error bars by multiple runs (**Table R1 and Table R2**);

W2: 10% poisoning ratio scenario for poisoning defense method (**Table R1**);

**Reviewer A3yL**

W1 & Q1: Experiments on other dataset, e.g., LAION (**Figure R2**);

**Reviewer jBz2**

W1 & W2 & Q2: Experiments’ error bars by multiple runs (**Table R1 and Table R2**);

Q1: Visualizations of G4 (**Figure R1 and Figure R2**);

**Reviewer HqnS**

W2: Experiments on other dataset, e.g., LAION (**Figure R2**);

W6: Experiments on the data replication, comparing images once poisoned vs. once not poisoned (**Figure R3**);

Q3 & Q4: Experiment on more trigger patterns (**Figure R1**).

---

### Decision · Program_Chairs · 2024-09-25

**Decision:**

Accept (poster)

**Comment:**

This paper studies data poisoning attacks on diffusion models. One of the main motivations is to attack diffusion models with only access to the training dataset, as opposed to also the objectives and sampling procedures. The authors show several findings regarding model effects they could influence using their attack method. One is that the generated images could either be misaligned with the input prompt or the images could match the prompt while containing certain correlated content (called triggers in this work). Another outcome of this work is to defend against poisoning attacks, including detecting poisoned training data or enhancing robustness against poisoning attacks. Overall, the paper has both high level messages (regarding the ease of poisoning diffusion models) as well as technical contributions (empirically showing how to attack/defend in the space of image generation).

During the rebuttal phase, the authors addressed many of the key issues that the reviewers raised. For example, the authors provide a small experiment showing that their method can work on data without class labels, using captions as a proxy instead. The authors also provide error bars and other investigations into the reproducibility of their work. Moreover, after the rebuttal, many of the reviewers have raised their scores. After the author-reviewer discussion, the four reviewers are unanimous in the support and merits of the submitted work, without any major criticisms. In total, all four reviewers now vote for acceptance of the paper, which is a rare outcome in high level ML conferences.

Given the soundness and thoroughness of the presented work, I vote for acceptance.

I encourage the authors to fully revise the manuscript for the final version of the paper, fixing typos, adding references, and including the additional experiments (e.g., the LAION experiments and the error bars). In my opinion, extending the attack to text-image data (w/o class labels) is natural and interesting, and it would be worth scaling up this experiment to more than 500 images for the final version of the paper (as suggested in the reviewer responses). Please also enlarge the figures in the appendix of the paper so that it is easier to see the images and so that the plots with overlayed text have better spacing and are less crowded. There is no page limit for the appendix, so it is ok to have larger figures that are more clear.